# Unlocking the Sugar Code: Implications and Consequences of Glycosylation in Alzheimer’s Disease and Other Tauopathies

**DOI:** 10.3390/biomedicines13122884

**Published:** 2025-11-26

**Authors:** Andrei-Cristian Bondar, Marius P. Iordache, Mirela Coroescu, Anca Buliman, Elena Rusu, Magdalena Budișteanu, Cristiana Tanase

**Affiliations:** 1Faculty of Medicine, Titu Maiorescu University, 040441 Bucharest, Romania; andrei.bondar@prof.utm.ro (A.-C.B.); coroescumirela@yahoo.com (M.C.); buliman_anca@yahoo.com (A.B.); elenarusu98@yahoo.com (E.R.); magda_efrim@yahoo.com (M.B.); cristianatp@yahoo.com (C.T.); 2Psychiatry Hospital “Prof. Dr. Alexandru Obregia”, Soseaua Berceni 10, 041914 Bucharest, Romania; 3Ilfov County Clinical Emergency Hospital, 022104 Bucharest, Romania; 4“Nicolae Cajal” Medical Institute, Faculty of Medicine and Pharmacy, Titu Maiorescu University, 011413 Bucharest, Romania; 5“Victor Babeș” National Institute of Pathology, 050096 Bucharest, Romania

**Keywords:** Alzheimer’s disease, tauopathies, glycosylation, *N*-glycosylation, *O*-glycosylation, amyloid-β, tau protein, neurodegeneration, post-translational modifications, glycoproteomics

## Abstract

Alzheimer’s disease (AD) is the most prevalent cause of dementia, characterized by progressive cognitive decline, amyloid-β (Aβ) plaques, and neurofibrillary tangles composed of hyperphosphorylated tau protein. Other tauopathies, including frontotemporal lobar degeneration (FTLD), progressive supranuclear palsy (PSP), and corticobasal degeneration (CBD) share pathological hallmarks centered on abnormal tau biology. Increasing evidence highlights the role of post-translational modifications in modulating these pathogenic processes. Among these, glycosylation, the enzymatic attachment of glycans to proteins or lipids, has emerged as a critical regulator of protein folding, trafficking, aggregation, and clearance. Both *N*-linked glycosylation (*N*-glycosylation) and *O*-linked glycosylation (*O*-glycosylation) influence tau stability, Aβ processing, receptor signaling, synaptic integrity, and neuroinflammation. Dysregulated glycosylation patterns have been documented in brains and cerebrospinal fluid (CSF) of AD patients, suggesting biomarker potential and novel therapeutic targets. Moreover, glycosyltransferases and glycosidases show altered expression in neurodegeneration, linking metabolic and inflammatory pathways to tauopathy progression. This review synthesizes current evidence on the implications and consequences of glycosylation in AD and other tauopathies, integrating mechanistic, pathological, and clinical findings. We also discuss advances in glycoproteomics, the interplay between glycosylation and phosphorylation, and the translational potential of targeting glycosylation pathways for diagnosis and therapy.

## 1. Introduction

Alzheimer’s disease (AD) is the leading cause of dementia affecting over 50 million individuals worldwide, with prevalence projected to rise sharply as populations age. Clinically, AD is defined by progressive cognitive decline, memory impairment, and eventual loss of autonomy [1]. Neuropathologically, it is characterized by extracellular amyloid-β (Aβ) plaques and intracellular neurofibrillary tangles (NFTs) composed of hyperphosphorylated tau protein [2]. While these lesions form the cornerstone of the amyloid and tau hypotheses, AD is increasingly recognized as a multifactorial disorder, integrating immune, vascular, metabolic, and protein homeostasis pathways.

A crucial layer of regulation in these processes is post-translational modification (PTM), which fine-tunes protein structure, stability, and localization. Among PTMs, tau phosphorylation has been extensively studied, but glycosylation has gained momentum as a determinant of disease progression [3]. Glycosylation refers to the enzymatic addition of glycans to proteins or lipids. Two major classes relevant to AD and tauopathies are *N*-glycosylation, where glycans attach to asparagine residues and *O*-glycosylation, where glycans attach to serine or threonine residues [4].

Variants such as *O*-GlcNAcylation—the dynamic attachment of *N*-acetylglucosamine—play pivotal roles in regulating protein interactions and preventing aggregation. Glycosylation influences protein folding, trafficking, degradation, and cell–cell signaling, processes that are fundamental to neuronal viability [5].

Tauopathies, a group of neurodegenerative diseases characterized by abnormal aggregation of the microtubule-associated protein tau include AD, progressive supranuclear palsy (PSP), corticobasal degeneration (CBD), and Pick’s disease. These disorders share tau accumulation but differ in isoform composition, filament morphology, and regional pathology [4,5].

In tauopathies such as frontotemporal lobar degeneration (FTLD), PSP, and CBD, glycosylation abnormalities intersect with tau phosphorylation, altering aggregation dynamics and neuronal toxicity [4]. Similarly, amyloid precursor protein (APP) processing and Aβ secretion are influenced by *N*-glycosylation, linking glycan biology to amyloid pathology [5]. Moreover, glycosylation affects receptor signaling, immune activation, and synaptic function, positioning it as a central regulator of neurodegeneration [3].

Advances in glycoproteomics now enable precise mapping of glycosylation changes in AD brain tissue, cerebrospinal fluid (CSF), and serum, revealing disease-specific signatures with potential diagnostic and prognostic value [1]. Furthermore, enzymes such as glycosyltransferases and glycosidases emerge as candidate drug targets while metabolic pathways regulating the hexosamine biosynthetic pathway (HBP) link systemic glucose handling to neuronal glycosylation patterns [4].

This review synthesizes current knowledge on the implications and consequences of glycosylation in AD and other tauopathies, with focus on tau biology, amyloid processing, immune and synaptic modulation, and translational potential for biomarker discovery and therapy.

## 2. Tau Glycosylation in Alzheimer’s Disease and Related Tauopathies

Tau protein is a microtubule-associated protein essential for stabilizing cytoskeletal architecture and supporting axonal transport [4,5]. In AD and other tauopathies, tau undergoes extensive PTMs that alter its biochemical and biophysical properties. While hyperphosphorylation has been studied extensively, evidence increasingly demonstrates that glycosylation is a critical determinant of tau structure, aggregation, and toxicity [6].

### 2.1. N-Glycosylation of Tau

Early studies revealed that tau is aberrantly *N*-glycosylated in AD brains, whereas normal adult tau is typically not glycosylated [7]. The addition of *N*-linked glycans occurs at asparagine residues and alters the conformation of tau, reducing its affinity for microtubules while increasing aggregation propensity [8]. *N*-glycosylated tau has been detected within paired helical filaments and neurofibrillary tangles, supporting a pathogenic role [4].

Importantly, aberrant *N*-glycosylation often precedes or facilitates hyperphosphorylation. Glycosylated tau shows increased susceptibility to kinases such as glycogen synthase kinase-3β (GSK3β), leading to pathogenic phosphorylation pattern. This sequence highlights glycosylation as an upstream event in tauopathy pathogenesis [7]. Glycosyltransferases—enzymes that catalyze the transfer of sugar moieties to proteins or lipids—mediate the covalent addition of glycan structures that can modulate tau folding, stability, and aggregation [6,7].

### 2.2. O-GlcNAcylation of Tau

Another critical modification is *O*-GlcNAcylation, the addition of *O*-linked *N*-acetylglucosamine to serine or threonine residues. Unlike *N*-glycosylation, *O*-GlcNAcylation appears protective. Increased *O*-GlcNAcylation reduces tau phosphorylation and aggregation by competing with phosphorylation sites [8]. Animal studies show that enhancing *O*-GlcNAcylation stabilizes tau, improves neuronal survival, and ameliorates memory deficits [5].

However, AD brains consistently show reduced *O*-GlcNAcylation, correlating with increased phosphorylation and tangle burden [9]. This reduction may result from impaired glucose metabolism and flux through the HBP [3].

### 2.3. Glycosylation in Other Tauopathies

In FTLD, PSP, and CBD, tau aggregates share common modifications. Abnormal glycosylation of tau has been demonstrated across these conditions, despite disease-specific differences in glycosylation site occupancy and glycan composition [10]. In PSP, tau shows increased high-mannose *N*-glycans, while in CBD, complex-type *N*-glycans are enriched, suggesting distinct enzymatic dysregulation [11].

Moreover, *O*-GlcNAcylation deficits are not restricted to AD but also appear in FTLD brain tissue, further linking impaired glucose metabolism to tau pathology across tauopathies [9].

While all tauopathies share abnormal tau aggregation, key differences exist in isoform composition, filament structure, and regional vulnerability. In AD, neurofibrillary tangles contain both 3-repeat (3R) and 4-repeat (4R) tau isoforms that assemble into paired helical filaments (PHFs), typically distributed from the entorhinal cortex to the neocortex in a hierarchical pattern [9]. In contrast, PSP and CBD predominantly accumulate 4R-tau, forming straight filaments and involving basal ganglia, brainstem, and motor cortex [8,9]. FTLD-tau subtypes vary—Pick’s disease features 3R-tau inclusions (Pick bodies) within frontotemporal regions, whereas globular glial tauopathies exhibit mixed isoforms with astrocytic or oligodendroglial predominance [10]. These distinctions suggest that the biochemical context of tau, including PTMs such as glycosylation, may differ by isoform and brain region. Evidence indicates that AD-associated tau tends to be more extensively *N*-glycosylated and exhibits reduced *O*-GlcNAcylation, while PSP and CBD display more restricted *N*-glycan complexity and relatively conserved *O*-GlcNAc profiles [11]. Thus, differential tau glycosylation may contribute to disease-specific aggregation patterns, cellular tropism, and progression rates among tauopathies [9,10,11].

### 2.4. Interplay Between Glycosylation and Phosphorylation

Perhaps the most critical insight is the interplay between glycosylation and phosphorylation. *N*-glycosylated tau is a better substrate for phosphorylation, while *O*-GlcNAcylation prevents phosphorylation at adjacent residues [12]. This creates a pathogenic imbalance, reduced *O*-GlcNAcylation and increased *N*-glycosylation synergistically drive tau hyperphosphorylation and aggregation [13].

This dynamic crosstalk suggests that therapies aimed at restoring *O*-GlcNAcylation or preventing aberrant *N*-glycosylation could rebalance tau homeostasis and slow disease progression. Indeed, small-molecule inhibitors of *O*-GlcNAcase (OGA), the enzyme removing *O*-GlcNAc, are under investigation as disease-modifying treatments [6].

### 2.5. Comparative Analysis of Tau Glycosylation and Phosphorylation Sites

*N*-glycosylation appears to promote tau hyperphosphorylation and aggregation, whereas *O*-GlcNAcylation exerts a protective role [11]. However, the relationship between glycosylation and phosphorylation is highly dependent on site specificity and structural context. Tau protein contains over 80 potential phosphorylation sites and at least 10 experimentally confirmed glycosylation sites, many of which are spatially proximal or overlapping [12]. *N*-glycosylation typically occurs on asparagine residues (e.g., Asn167, Asn359), whereas *O*-GlcNAcylation affects serine and threonine residues (e.g., Ser400, Thr123, Thr245, Ser356) that often coincide with major phosphorylation motifs recognized by kinases such as GSK3β, cyclin-dependent kinase 5 (CDK5), and microtubule affinity-regulating kinase 2 (MARK2) [13]. These modifications act in a competitive and regulatory manner. *N*-glycosylation changes the tertiary structure of tau, reducing its microtubule affinity and exposing neighboring serine and threonine residues to kinases, thereby facilitating hyperphosphorylation. In contrast, *O*-GlcNAcylation directly competes with phosphorylation at the same or adjacent residues, inhibiting excessive phosphorylation and stabilizing tau–microtubule interactions [12]. Consequently, decreased *O*-GlcNAcylation in AD—often linked to impaired glucose metabolism—removes this protective brake, leading to pathological phosphorylation and tangle formation [11,12,13]. In AD, both 3R and 4R tau isoforms exhibit extensive *N*-glycosylation and reduced *O*-GlcNAcylation, driving PHF assembly [10]. In other tauopathies such as PSP and CBD, tau aggregates are composed predominantly of 4R isoforms with more restricted glycosylation patterns and distinct phosphorylation site usage (e.g., Ser262, Ser396) [10,11]. These biochemical differences suggest that glycosylation and phosphorylation site occupancy contribute to the diversity of filament morphology and regional vulnerability observed across tauopathies [10,11,13]. Future work should address these site-specific interactions, emphasizing that *N*-glycosylation (Asn167/359) acts as an early structural modifier that primes tau for pathogenic phosphorylation, while *O*-GlcNAcylation (Ser400, Thr245, Ser356) counteracts aggregation by masking phosphorylation motifs [11,13]. This comparative framework could enhance understanding of tau pathology across AD and non-AD tauopathies and guide the design of site-directed therapeutic interventions.

## 3. Glycosylation and Amyloid-β Pathology

The amyloid cascade hypothesis proposes that abnormal processing of APP initiates a series of pathogenic events culminating in AD [8]. Glycosylation profoundly influences APP trafficking, secretase accessibility, and Aβ generation [5].

### 3.1. Amyloid Precursor Protein N-Glycosylation and Trafficking

APP is a heavily *N*-glycosylated type I transmembrane protein. *N*-glycans at asparagine residues near its luminal domain regulate APP folding, trafficking through the secretory pathway, and stability at the cell surface [14]. Disruption of these *N*-glycans alters APP sorting, diverting it toward endosomal compartments enriched in β-site β-secretase enzyme, APP-cleaving enzyme 1 (BACE1), thereby increasing amyloidogenic cleavage [15].

Experimental studies confirm that loss of APP *N*-glycosylation enhances Aβ production, whereas stabilizing *N*-glycan structures reduces amyloidogenic processing [16]. These findings indicate that altered *N*-glycosylation in the AD brains may bias APP processing toward Aβ generation.

### 3.2. O-Glycosylation and Secretase Regulation

In addition to *N*-glycosylation, APP and its secretases undergo *O*-glycosylation, particularly *O*-GalNAc modification, which influences protein trafficking and enzymatic activity. *O*-glycosylation of APP modulates its endocytosis rate, thereby controlling access to β- and γ-secretases [5]. Reduced *O*-glycosylation correlates with increased amyloidogenic cleavage and elevated extracellular Aβ accumulation [8].

Secretases themselves are glycoproteins. BACE1 contains multiple *N*-glycans that regulate folding, transport, and stability. Inhibition of BACE1 glycosylation reduces its enzymatic activity and impairs Aβ generation [17]. Similarly, γ-secretase components such as nicastrin are highly glycosylated. The glycan structures near the substrate-binding pocket determine APP cleavage patterns. These modifications fine-tune Aβ species ratios, including the pathogenic Aβ42/40 balance [18].

### 3.3. Aberrant Glycosylation of Amyloid-β Peptides

Emerging evidence suggests that Aβ peptides themselves can undergo glycosylation or interact with glycans. Modified Aβ shows altered aggregation kinetics and toxicity. Glycosylated Aβ peptides aggregate faster and form more neurotoxic oligomers compared to unmodified Aβ [6]. Additionally, glycans on neuronal membranes interact with Aβ, promoting deposition and impairing clearance [1].

Glycosylation also modulates Aβ clearance via receptor-mediated endocytosis. For instance, the receptor for advanced glycation end products (RAGE) binds both glycated proteins and Aβ, facilitating their transcytosis across the blood–brain barrier [2]. Enhanced RAGE glycosylation increases its affinity for Aβ, promoting deposition in the brain parenchyma [19].

The interplay between glycosylation and Aβ toxicity extends beyond processing and clearance. Aberrant glycosylation of synaptic proteins, receptors, and ion channels exacerbates Aβ-induced dysfunction. For example, *N*-methyl-*D*-aspartate (NMDA) receptors require *N*-glycosylation for proper surface expression as loss of this modification sensitizes neurons to Aβ-mediated excitotoxicity [20]. Similarly, glycosylation of prion protein (PrP^C^) modulates its interaction with Aβ oligomers, influencing synaptotoxic signaling cascades [17,21].

Furthermore, microglial and astrocytic receptors involved in Aβ clearance—such as triggering receptor expressed on myeloid cells 2 (TREM2) and cluster of differentiation 33 (CD33)—are heavily glycosylated. Altered glycan structures modulate receptor affinity and downstream inflammatory responses [22]. This suggests that glycosylation contributes to the balance between protective clearance and harmful neuroinflammation.

### 3.4. Implications for Therapy

Given the centrality of glycosylation in APP metabolism and Aβ biology, targeting glycosylation pathways represents a promising therapeutic approach. Experimental strategies include inhibiting BACE1 glycosylation, enhancing protective *O*-glycosylation, and modulating glycosyltransferases that remodel glycans on APP and its processing enzymes [16,17]. Additionally, blocking RAGE glycosylation has been proposed to limit Aβ transport into the brain and reduce Aβ deposition within the cerebral parenchyma and vasculature [2,19,23,24,25].

## 4. Glycosylation, Synaptic Function, and Neuroinflammation

Synaptic dysfunction and chronic neuroinflammation are central drivers of neurodegeneration in AD and tauopathies. Both processes are intimately modulated by glycosylation, which shapes receptor signaling, immune cell activation, and glial responses [22,26,27].

### 4.1. Glycosylation of Synaptic Receptors and Adhesion Molecules

Neuronal communication depends on the proper glycosylation of neurotransmitter receptors and cell adhesion proteins. NMDA receptors, critical for synaptic plasticity, require *N*-glycosylation for assembly, trafficking, and surface expression. In AD models, aberrant glycosylation disrupts NMDA receptor localization, sensitizing neurons to Aβ-induced excitotoxicity [20].

Similarly, α-amino-3-hydroxy-5-methyl-4-isoxazolepropionic acid (AMPA) receptors rely on *N*-glycans for stability and gating properties. Altered glycosylation reduces synaptic strength and long-term potentiation, processes essential for learning and memory [15].

Adhesion molecules such as neural cell adhesion molecule (NCAM) and L1CAM are highly glycosylated, and polysialylation of NCAM regulates neurite outgrowth and synaptic remodeling. In addition, reduced polysialylation has been detected in AD hippocampus, correlating with impaired synaptic connectivity [28].

### 4.2. Immune Receptor Glycosylation and Microglial Activation

Microglia, the brain’s resident immune cells, rely on glycosylated receptors for recognition and clearance of pathological proteins [22]. TREM2 contains multiple *N*-glycosylation sites that regulate folding and surface expression [26]. Mutations altering TREM2 glycosylation reduce receptor stability, impair microglial phagocytosis of Aβ, and increase AD risk [22,26,27].

Another example is CD33, a sialic acid-binding receptor implicated in AD genetic risk [6,8,19]. CD33 glycosylation governs ligand binding and inhibitory signaling while aberrant glycosylation enhances its inhibitory effect, reducing Aβ clearance and exacerbating inflammation [12].

Astrocytes also display glycosylated receptors that influence neuroinflammation. For instance, glial fibrillary acidic protein (GFAP)-interacting proteins undergo glycan remodeling in reactive astrocytes, altering cytokine release and glial scar formation [5,29].

### 4.3. Cytokines, Chemokines, and Glycosylation

Inflammatory mediators in the central nervous system (CNS) are heavily glycosylated. Interleukin-6 (IL-6), a pro-inflammatory cytokine elevated in AD, requires *N*-glycosylation for secretion and receptor binding [29]. Altered IL-6 glycosylation patterns modulate its bioactivity and half-life in the extracellular space [29,30,31]. Similarly, tumor necrosis factor-alpha (TNF-α) and interleukin-1β (IL-1β) undergo glycosylation changes in AD, influencing receptor affinity and downstream signaling [32,33,34].

Chemokine receptors such as CCR5 and CX3CR1 are also glycosylated, with changes in glycan branching modulating microglial migration and response to injury [11]. In AD, aberrant glycosylation of these receptors contributes to chronic non-resolving inflammation [11,28].

### 4.4. Glycosylation and the Complement System

The complement cascade, a major mediator of synaptic pruning and neuroinflammation, is highly influenced by glycosylation [7]. C1q, the initiator of the classical pathway, carries sialylated glycans that regulate binding to immune complexes and synaptic elements [7,11]. Loss of sialylation enhances complement activation and drives synapse elimination in AD models [9,35].

Other complement proteins, including C3 and factor H, show altered glycosylation in AD brain and CSF, correlating with disease severity [5,8,14,23,26]. This suggests that glycosylation defects amplify maladaptive immune responses.

### 4.5. Neuroinflammation Beyond Amyloid and Tau

Glycosylation also extends its influence to pattern recognition receptors (PRRs) that detect damage-associated molecular patterns (DAMPs) [2]. RAGE is heavily glycosylated, and its modification enhances binding to glycated proteins and Aβ, facilitating inflammatory cascades [2,26]. Glycosylated RAGE amplifies NF-κB signaling, perpetuating neuroinflammation [3].

In addition, toll-like receptors (TLRs) rely on *N*-glycans for stability and ligand recognition [35]. Dysregulated TLR glycosylation biases microglia toward pro-inflammatory phenotypes, further fueling pathology [35,36,37,38,39].

### 4.6. Therapeutic Implications

Given its central role in receptor biology and inflammatory signaling, modulating glycosylation could rebalance immune responses. Potential strategies include glycoengineering of anti-inflammatory cytokines, inhibition of aberrant receptor glycosylation, or restoring protective glycosylation patterns on complement regulators [2,12,40].

## 5. Advances in Glycoproteomics and Biomarker Discovery

Biomarker discovery remains a critical frontier in AD and tauopathies. While classical markers such as CSF amyloid-β42 (Aβ42), total tau (t-tau), and phosphorylated tau (p-tau) have clinical utility, their specificity and predictive power are limited [32]. Glycoproteomics, the large-scale study of glycosylated proteins, has emerged as a powerful approach to identify novel biomarkers by leveraging disease-specific glycosylation changes [34,41,42,43]. Glycoproteomics provides detailed mapping of glycan sites and structures, enabling the identification of disease-specific tau modifications [34].

### 5.1. Glycoproteomic Alterations in Alzheimer’s Disease

Proteomic studies demonstrate profound changes in glycosylation within AD brains. Comparative analyses of postmortem cortical tissue show altered *N*-glycan branching, sialylation, and fucosylation across multiple glycoproteins [1,44,45]. Specific glycosylation signatures distinguish AD from age-matched controls, implicating disrupted glycosyltransferase and glycosidase activity [3,33].

In CSF, glycoproteomics reveals increased bisected *N*-glycans and altered glycoforms of tau and APP fragments, correlating with disease severity [46,47]. Serum analyses have also detected disease-specific glycosylation patterns, including reduced sialylation of acute-phase proteins and altered IgG Fc (Immunoglobulin G—fragment crystallizable) glycosylation, linking systemic inflammation to neurodegeneration [10,14,24,32,47,48].

### 5.2. Glycosylation as a Diagnostic and Prognostic Biomarker

The diagnostic potential of glycosylation is underscored by studies demonstrating that altered glycosylation of tau precedes overt phosphorylation changes [49]. Detection of glycosylated tau in CSF or plasma could thus provide early disease biomarkers. Likewise, *O*-GlcNAcylation status of tau correlates inversely with disease progression, suggesting its utility as a prognostic marker [46,50,51].

Moreover, glycan-based profiling has shown promise in differentiating AD from other dementias. For instance, patients with FTLD and PSP display distinct glycosylation patterns compared to AD, enabling differential diagnosis [16,20].

### 5.3. Mass Spectrometry and Glycoproteomic Technologies

Recent advances in MS have revolutionized glycoproteomics. Methods such as electron-transfer dissociation (ETD) and higher-energy collisional dissociation (HCD) provide site-specific glycan characterization with high sensitivity [42]. Enrichment strategies using lectins, hydrophilic interaction chromatography (HILIC), and nanoporous materials further enhance glycopeptide detection [5,12,52].

Novel probes and nanocomposites allow in situ visualization of glycosylation in brain tissue, thereby bridging biochemical analyses with histopathology. These tools have revealed regional differences in glycosylation within hippocampus, cortex, and subcortical structures, aligning with clinical phenotypes of memory loss, executive dysfunction, and motor impairment [53,54].

Given the complexity of tau PTMs, including phosphorylation and glycosylation, the development of specific and validated antibodies remains a significant challenge. Most commercially available anti-tau antibodies are designed against phosphorylated epitopes (e.g., pSer202, pThr231, pSer396), many of which have been extensively validated for immunohistochemistry (IHC), Western blot, and enzyme-linked immunosorbent assay (ELISA) applications in brain and CSF samples [12]. In contrast, antibodies specifically targeting glycosylated tau are scarce and often lack full validation due to the structural diversity and low abundance of glycoforms [53]. Only a limited number of monoclonal antibodies raised against *N*-glycosylated or *O*-GlcNAc-modified tau have been reported, with most data derived from research-grade reagents used in immunoprecipitation or lectin-based assays rather than standardized diagnostic kits [12,53].

Validation of tau glycosylation-specific antibodies for clinical biofluids such as CSF or serum remains preliminary, primarily constrained by low antigen concentration and conformational masking of glycan epitopes [52,53]. While recent studies have employed antibody–lectin hybrid assays and antibody enrichment followed by liquid chromatography (LC)–MS/MS to enhance specificity, no single antibody currently demonstrates broad cross-platform reliability [5,12]. For IHC detection, phospho-tau antibodies remain highly reliable and routinely used to stage pathology (e.g., AT8, PHF-1, CP13), whereas glycosylation-targeting antibodies are still in experimental phases [54]. As glycoproteomic characterization advances, development of rigorously validated monoclonal antibodies recognizing specific glycoepitopes on tau will be critical for translating glycosylation research into clinical biomarker applications [5,12,52,53,54].

### 5.4. Immunoglobulin Glycosylation and Systemic Biomarkers

Beyond CNS, peripheral immune system glycosylation offers insights into neurodegeneration. Altered IgG Fc glycosylation has been documented in AD, with reduced galactosylation and sialylation associated with pro-inflammatory phenotypes [20,47,55]. These changes may reflect systemic inflammation or feedback from neuroinflammatory cascades.

Additionally, plasma glycoproteomic profiling identified altered glycosylation of acute-phase reactants such as haptoglobin and α1-acid glycoprotein, linking metabolic stress to disease progression [56,57,58].

### 5.5. Integration with Multimodal Biomarkers

Glycosylation-based biomarkers should not be considered in isolation but integrated with imaging, fluid, and genetic markers. Combining CSF tau glycosylation profiles with positron emission tomography (PET) amyloid imaging improves diagnostic accuracy over either modality alone [10,51]. Similarly, serum glycoproteomic signatures complement plasma phosphorylated tau (phospho-tau181, phospho-tau217) measurements, enhancing early detection potential [11,46].

### 5.6. Challenges and Opportunities

Despite major progress, glycoproteomics faces challenges. Glycan heterogeneity complicates quantification and standardized reference libraries are limited [59]. Moreover, interindividual variability in glycosylation due to genetics, diet, and comorbidities complicates biomarker validation [1,59,60,61].

However, these same features offer opportunities. Disease-specific glycan “fingerprints” could yield highly specific biomarkers and integration with machine learning approaches promises improved classification [3,62,63]. In addition, longitudinal studies are beginning to reveal how glycosylation changes track disease trajectory, paving the way for prognostic applications [62].

Table 1 and Table 2 present analytical limitations in tau glycoproteomics and emerging solutions.

## 6. Enzymatic Regulators of Glycosylation in Neurodegeneration

The balance of protein glycosylation is tightly regulated by glycosyltransferases and glycosidases, as well as by metabolic flux through the HBP, which generates nucleotide sugars as glycosylation substrates. In Alzheimer’s disease and other tauopathies, dysregulation of these enzymes and pathways alters glycosylation homeostasis, thereby influencing APP metabolism, tau aggregation, receptor signaling, and neuroinflammation [1,50,57].

### 6.1. Glycosyltransferases in AD and Tauopathies

*N*-acetylglucosaminyltransferases (GnTs) regulate branching and extension of *N*-glycans. Upregulation of GnT-III increases bisected *N*-glycans on tau and APP, promoting pathological processing [16]. Similarly, elevated GnT-V activity has been detected in AD brains, producing β1,6-GlcNAc branching that alters synaptic receptor function [32,64,65].

Sialyltransferases, which attach sialic acid residues, are dysregulated in AD, leading to reduced sialylation of neuronal adhesion molecules such as NCAM and impaired synaptic remodeling [18]. Altered sialylation also contributes to immune dysfunction by modifying microglial receptor interactions with sialylated ligands [36,66].

In tauopathies beyond AD, such as PSP and CBD, distinct changes in glycosyltransferase expression drive unique glycosylation signatures. Comparative analyses reveal elevated fucosyltransferase activity in PSP, whereas CBD brains show increased sialyltransferase expression, contributing to disease-specific tau glycoforms [11,16,67].

### 6.2. Glycosidases and Tau Pathology

Glycosidases remove glycans from proteins and modulate glycan turnover. In AD, elevated β-*N*-acetylglucosaminidase (β-*N*-OGA) activity reduces protective *O*-GlcNAcylation of tau, facilitating its hyperphosphorylation and aggregation [20,59,68]. Conversely, inhibition of β-*N*-OGA increases *O*-GlcNAcylation and reduces neurodegeneration in tauopathy mouse models [69,70].

Neuraminidases (sialidases) regulate the removal of sialic acids. Increased neuraminidase activity has been linked to loss of polysialylation on NCAM in AD hippocampus, impairing synaptic plasticity [18,32,71]. Additionally, altered lysosomal glycosidase activity disrupts degradation of glycoproteins, contributing to protein aggregation and lysosomal storage pathology observed in AD [3]. Abnormal *N*-glycosylation of tau promotes hyperphosphorylation and aggregation, while reduced *O*-GlcNAcylation removes a protective brake on phosphorylation. *N*- and *O*-glycosylation of APP and secretases regulate Aβ generation, with aberrant patterns favoring amyloidogenic processing [16,71]. Synaptic receptors and adhesion molecules require proper glycosylation for stability and plasticity [72]. Their dysregulation contributes to synaptic failure [73]. Immune receptors (TREM2, CD33, and RAGE) and cytokines are glycosylated, shaping microglial and astrocytic activation, while complement protein glycosylation regulates synapse pruning. Finally, altered activity of glycosyltransferases, glycosidases, and the HBP underlies these shifts. Together, these processes highlight glycosylation as both a pathogenic mechanism and therapeutic target [72,73]. A schematic overview of glycosylation is presented in Figure 1.

### 6.3. Hexosamine Biosynthetic Pathway and Metabolic Regulation

HBP provides UDP-*N*-acetylglucosamine (UDP-GlcNAc), the substrate for *O*-GlcNAcylation [60]. Reduced glucose flux into the HBP, due to impaired brain glucose metabolism in AD, diminishes *O*-GlcNAcylation levels [60,72,73]. This metabolic defect links systemic insulin resistance and type 2 diabetes to enhanced risk of AD and tauopathies [74,75,76].

Experimental restoration of HBP flux via glucosamine supplementation increases *O*-GlcNAcylation and reduces tau phosphorylation in animal models, highlighting its therapeutic potential [68]. Similarly, caloric restriction and metabolic interventions that enhance glucose utilization may indirectly restore protective glycosylation patterns [68,77,78].

### 6.4. Crosstalk with Phosphorylation Pathways

The interplay between glycosylation and phosphorylation is tightly controlled by enzymes. Reduced *O*-GlcNAc transferase (OGT) activity decreases tau *O*-GlcNAcylation, thereby exposing phosphorylation sites for kinases such as GSK3β [30,57,79,80]. Conversely, enhancing OGT activity shifts the balance toward protective *O*-GlcNAcylation [6,56].

This enzymatic tug-of-war suggests that targeting glycosylation enzymes may indirectly modulate tau phosphorylation, providing a dual mechanism for therapeutic intervention [34,68,81].

### 6.5. Enzymatic Dysregulation as Biomarkers

Aberrant expression of glycosylation enzymes is detectable in CSF and serum, offering biomarker potential [3]. Elevated OGA and reduced OGT levels correlate with higher CSF tau and worse cognitive outcomes [3,29,82,83]. Similarly, altered glycosyltransferase expression patterns in peripheral blood mononuclear cells reflect disease stage and may serve as accessible biomarkers [39,76].

### 6.6. Clinical Correlations of Tau Glycosylation with Cognitive Decline and Disease Staging

Emerging evidence indicates that aberrant tau glycosylation patterns are not only neuropathological hallmarks but may also correlate with measurable clinical outcomes. In AD, elevated *N*-glycosylated tau in brain tissue and CSF has been associated with advanced Braak stages and greater neurofibrillary tangle burden, reflecting a molecular shift toward aggregation-prone conformers [29,39]. The Braak staging system assigns cases to six stages based on the presence of Lewy pathology in the medulla oblongata (I), pons (II), mesencephalon (III), amygdala and other limbic structures (IV), association neocortex (V), and first order sensory association neocortex (VI). Conversely, reduced *O*-GlcNAcylation levels in tau correlate with accelerated cognitive decline and lower Mini-Mental State Examination (MMSE) scores, suggesting a loss of neuroprotective regulation [76]. Longitudinal cohort studies have shown that specific CSF glycoforms of tau—particularly those carrying high-mannose *N*-glycans or lacking *O*-GlcNAc at Ser400 and Thr245—predict progression from mild cognitive impairment (MCI) to Alzheimer’s dementia over 2–4 years [76,82]. Furthermore, glyco-profiling in CSF and plasma using lectin-based assays or glycoproteomic MS has demonstrated potential for distinguishing AD from other tauopathies such as PSP and CBD, based on differences in glycan branching and sialylation patterns [82], as shown in Table 3. Collectively, these findings suggest that the glycosylation state of tau serves as a dynamic biochemical correlate of disease stage and may complement phosphorylated tau and Aβ42 as part of a next-generation biomarker panel for early detection and disease monitoring [83].

### 6.7. Therapeutic Implications

Enzymes regulating glycosylation are increasingly viewed as druggable targets. OGA inhibitors, such as thiamet-G, are under preclinical investigation for enhancing tau *O*-GlcNAcylation [39,57,66,84]. By preserving polysialylation, sialidase inhibitors may protect synaptic plasticity [81]. Small molecules targeting glycosyltransferases are less developed but hold potential for rebalancing glycosylation networks in neurodegeneration [8,81,85,86,87]. The glycosylation processes involved in AD and other tauopathies are summarized in Table 4.

## 7. Discussion

Growing evidence demonstrates the role of glycosylation as a central, driver of AD and related tauopathies. Unlike phosphorylation, glycosylation provides a multidimensional regulatory layer that influences protein stability, trafficking, aggregation, and clearance [1,19,89].

### 7.1. Glycosylation and the Hierarchy of Pathological Events

One of the persistent debates in AD research concerns the sequence of pathological events. To this end, glycosylation research offers new perspectives. Evidence suggests that *N*-glycosylation of tau occurs early, priming the protein for hyperphosphorylation and aggregation [64,66,90]. In parallel, aberrant glycosylation of APP biases processing toward amyloidogenic cleavage [5,91]. Thus, glycosylation abnormalities may precede both amyloid and tau pathologies, positioning them as an upstream “common denominator”.

### 7.2. Protective Versus Pathogenic Roles

Not all glycosylation is deleterious. *O*-GlcNAcylation of tau exemplifies a protective modification by competing with phosphorylation sites, which reduces aggregation and toxicity [16,77,92,93]. Conversely, reduced *O*-GlcNAcylation in AD brains correlates with worsened pathology [57,77,94]. Similarly, polysialylation of NCAM supports synaptic plasticity, but its reduction contributes to connectivity deficits [64,65,95,96]. The dualistic nature of glycosylation complicates therapeutic strategies. Hence, interventions must distinguish between protective and pathogenic glycosylation events [22,97].

### 7.3. Crosstalk with Metabolism and Phosphorylation

A recurring theme is the integration of glycosylation with broader cellular networks. Metabolic dysfunction, particularly impaired glucose utilization, reduces flux through the HBP, limiting substrate availability for *O*-GlcNAcylation [3,98,99]. This explains why AD brains show reduced *O*-GlcNAc despite increased phosphorylation pressure. Moreover, enzymatic regulators such as OGT and OGA directly shape the phosphorylation landscape of tau by modulating glycosylation at competing sites [20,79,100], as presented in Figure 2.

### 7.4. Neuroinflammation as a Glycosylation-Driven Amplifier

Neuroinflammation plays a key role in the neurodegenerative process. Glycosylation critically modulates immune receptor signaling [27]. TREM2 requires *N*-glycosylation for stability. Mutations impairing this modification reduce microglial Aβ clearance [27,101,102]. Conversely, aberrant glycosylation of CD33 enhances its inhibitory signaling, further dampening microglial phagocytosis [66,103]. Glycosylation of complement proteins amplifies maladaptive synaptic pruning [9,104]. Collectively, these findings highlight glycosylation as a molecular amplifier of neuroinflammatory cascades.

### 7.5. Biomarker Potential and Translational Challenges

Glycoproteomics offers promising avenues for biomarker discovery. Altered glycosylation of tau, APP, and inflammatory proteins has been consistently detected in CSF and serum [42,105,106,107]. These patterns not only differentiate AD from controls but may also discriminate between tauopathies such as PSP and CBD [67,108]. However, technical barriers remain. Glycan heterogeneity complicates quantification and interindividual variability poses validation challenges [1,109].

### 7.6. Therapeutic Perspectives

Therapeutic manipulation of glycosylation is a rapidly advancing field. OGA inhibitors that increase protective *O*-GlcNAcylation of tau are under active investigation [110]. Strategies targeting glycosyltransferases or glycosidases could theoretically rebalance glycosylation networks, though selectivity remains a major obstacle [5,16,111,112]. In addition, targeting glycosylation of immune receptors such as RAGE and TREM2 may offer immunomodulatory benefits [24,27,113].

### 7.7. Remaining Controversies

Despite major progress, several controversies remain. It is unclear whether glycosylation changes are primary drivers of pathology or secondary adaptations to cellular stress. Some argue that abnormal glycosylation results from metabolic and inflammatory dysfunction rather than initiating pathology [27,114]. Others suggest that glycosylation shifts act as compensatory mechanisms, initially protective but maladaptive when chronic [9,115]. Clarifying these temporal relationships will be essential for therapeutic translation.

Glycosylation has emerged as a pivotal post-translational modification in neurodegeneration, shaping the trajectory of pathology through its multifaceted impact on protein structure, stability, trafficking, and signaling. Across this review, the following themes consistently emerge. Aberrant *N*-glycosylation primes tau for hyperphosphorylation and aggregation while reduced *O*-GlcNAcylation removes a critical brake against pathogenic modifications [91,116]. *N*- and *O*-glycosylation of APP and its processing enzymes profoundly influence amyloidogenic cleavage, with downstream effects on Aβ burden [5,117]. Glycosylation of receptors and adhesion molecules determines synaptic resilience, while glycosylation of immune receptors (TREM2, CD33, and RAGE) amplifies maladaptive neuroinflammation [19,27,118]. Enzymes such as OGT, OGA, and glycosyltransferases orchestrate glycosylation balance. Their dysregulation, often linked to impaired glucose metabolism and HBP flux, couples systemic metabolic dysfunction to neurodegeneration [3,119]. Glycoproteomic studies reveal distinct glycosylation signatures in brain, CSF, and serum, offering diagnostic and prognostic markers and providing disease-specific fingerprints that differentiate AD from other tauopathies [11,34,120].

Collectively, glycosylation abnormalities are not epiphenomena but core contributors to disease mechanisms, interwoven with phosphorylation, inflammation, and metabolism [44,121,122].

### 7.8. Glycosylation-Focused Guidelines for Tauopathies: Toward Consensus on Tau Pathology

To help the field converge on comparable and clinically relevant measurements, we propose the following minimum reporting standards and decision rules for glycosylation in tauopathies, expanding beyond prior reviews by integrating site-resolved glycosylation with assay readiness and disease-specific signatures, as illustrated in Figure 3.

Table 5 presents recommended reporting standards, disease signatures, validation rules, and translation framework for tau glycoproteomics.

## 8. Future Directions

While significant progress has been made, several research priorities and therapeutic opportunities remain. Longitudinal human studies are needed to determine whether glycosylation changes precede, accompany, or follow amyloid and tau pathology [9,123]. This temporal clarity will inform whether glycosylation can serve as a true early biomarker or therapeutic entry point.

High-resolution glycoproteomics using advanced MS, lectin arrays, and nanoparticle probes should be applied to large, well-characterized cohorts [3]. Integration with imaging and fluid biomarkers will improve diagnostic specificity and enable personalized medicine [3,42,124]. Clinical translation of OGA inhibitors and other glycosylation-modulating compounds should be pursued cautiously with attention to the dualistic nature of glycosylation, protective vs. pathogenic [5]. Parallel approaches may include sialidase inhibitors to preserve polysialylation, or small-molecule modulators of glycosyltransferases [5,24,125].

Glycosylation does not act isolated. Integrating glycosylation with phosphoproteomics, metabolomics, and transcriptomics will reveal multi-layered disease networks [1,126,127]. Such approaches can identify convergent pathways and potential drug targets. Comparative studies across PSP, CBD, and FTLD will refine understanding of disease-specific glycosylation signatures and highlight both shared and divergent mechanisms [11,128,129,130]. This comparative lens will sharpen differential diagnosis and inform tailored interventions. Translation will require standardization of glycoproteomic assays, establishment of reference databases, and validation in multicenter cohorts [13,131,132,133]. Importantly, therapeutic interventions targeting glycosylation must demonstrate CNS specificity while minimizing systemic off-target effects [16,17,19,134,135,136,137,138,139,140,141,142].

Taken together, these findings position glycosylation at the crossroads of molecular pathology, systemic metabolism, and neuroimmune signaling in AD and related tauopathies. While the evidence underscores both pathogenic and protective aspects of glycosylation, it also highlights unresolved questions regarding timing, causality, and therapeutic feasibility [135,143]. The convergence of biochemical, glycoproteomic, and translational studies emphasizes that glycosylation is not a peripheral phenomenon but a central determinant of disease trajectory [133,144]. This integrative perspective sets the stage for the concluding remarks, which synthesize the mechanistic insights, biomarker potential, and therapeutic avenues into a broader framework for future research and clinical translation [11,105].

## 9. Conclusions

Glycosylation represents a critical but underappreciated frontier in neurodegeneration research. By modulating protein folding, phosphorylation, aggregation, immune activation, and synaptic function, it acts as both a pathogenic driver and protective modifier. Aberrant *N*-glycosylation of tau and APP promotes aggregation and amyloidogenic processing while reduced *O*-GlcNAcylation and polysialylation remove protective mechanisms. In parallel, altered glycosylation of receptors and inflammatory mediators amplifies neuroinflammation and accelerates disease progression.

These insights highlight glycosylation as both a biomarker source and a therapeutic target at the convergence of technological advances. Progress in glycoproteomics now allows detection of disease-specific glycan signatures in CSF and serum, supporting early diagnosis and differential classification among tauopathies. Therapeutic strategies aimed at rebalancing glycosylation, such as OGA inhibitors or glycosyltransferase modulators, are promising but require careful consideration of specificity and systemic effects.

Overall, by integrating glycosylation research with broader molecular and clinical frameworks of neurodegeneration, a new path can be opened toward tangible opportunities for biomarker development, earlier detection, improved prognostic tools, and innovative therapies.

## Figures and Tables

**Figure 1 biomedicines-13-02884-f001:**
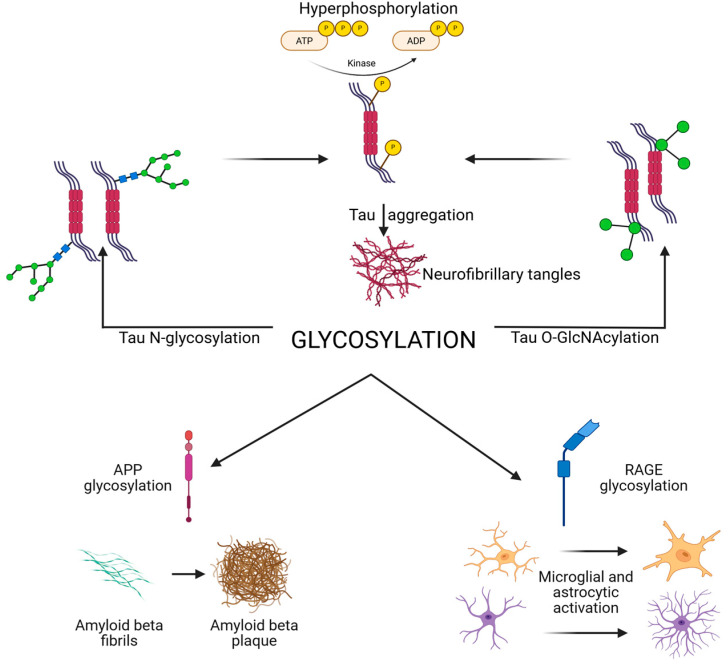
Schematic overview of glycosylation in AD and tauopathies, illustrating how glycosylation affects tau APP and Aβ, synaptic, and immune receptors, and the enzymatic and metabolic regulators (HBP, OGT, OGA, and GnTs). Created in BioRender. Iordache, M. (2025) https://BioRender.com/fbogzud (accessed on 18 October 2025). APP, amyloid precursor protein; HBP, hexosamine biosynthetic pathway; *N*-acetylglucosaminyltransferases; OGA, *O*-GlcNAcase; OGT, *O*-GlcNAc transferase; RAGE, receptor for advanced glycation end products.

**Figure 2 biomedicines-13-02884-f002:**
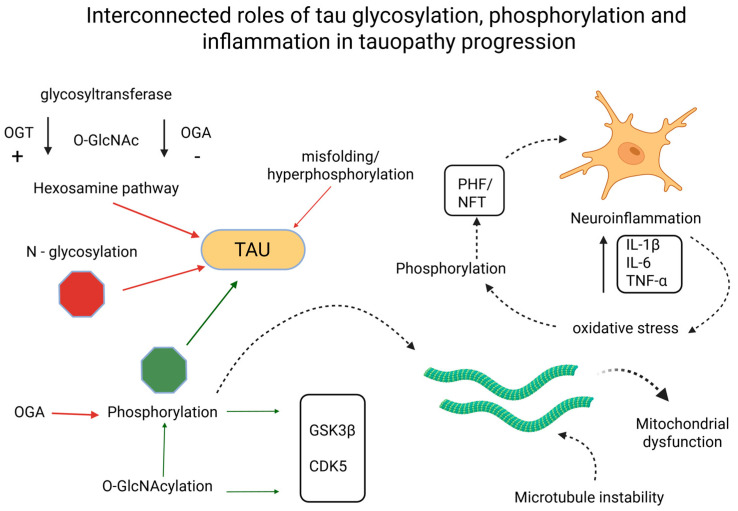
Interconnected roles of tau glycosylation, phosphorylation, and inflammation in tauopathy progression. Created in BioRender. Iordache, M. (2025) https://BioRender.com/psx2nng (accessed on 18 October 2025). CDK5, cyclin-dependent kinase 5; GSK3β, glycogen synthase kinase 3 beta; IL-1β, interleukin 1β; IL-6, interleukin 6; NFT, neurofibrillary tangle; OGA, *O*-GlcNAcase; OGT, *O*-GlcNAc transferase; *O*-GlcNAc, *O*-linked *N*-acetylglucosamine; *N*-glycosylation, asparagine-linked glycosylation; PHF, paired helical filament; TNF-α, tumor necrosis factor α.

**Figure 3 biomedicines-13-02884-f003:**
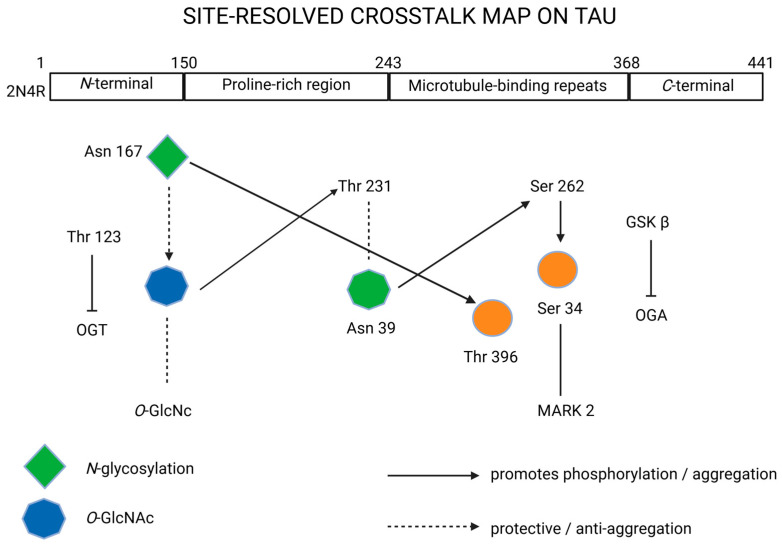
Site-resolved crosstalk on tau protein. Created in BioRender. Iordache, M. (2025) https://BioRender.com/zq3mkae (accessed on 18 October 2025). 2N4R, tau isoform with 2 *N*-terminal inserts and 4 microtubule-binding repeats; Asn, asparagine; *C*-terminal, carboxy-terminal region; GSKβ, glycogen synthase kinase 3-β; MARK2, microtubule affinity-regulating kinase 2; *N*-glycosylation, *N*-linked glycosylation; *N*-terminal, *N*-terminal region; OGA, *O*-GlcNAcase; *O*-GlcNAc, *O*-linked *N*-acetylglucosamine; OGT, *O*-GlcNAc transferase; Ser, serine; Thr, threonine.

**Table 1 biomedicines-13-02884-t001:** Analytical limitations in tau glycoproteomics.

Analytical Limitation	Description	References
1. Glycoform and glycan isomer complexity	Single glycosites carry multiple glycoforms; many glycans are isomeric, complicating confident assignment.	[1]
2. Fragmentation limitations	Collision-based fragmentation preferentially cleaves labile glycosidic bonds → good glycan info but poor peptide backbone coverage; limits site localization on multiply modified peptides.	[3]
3. Enrichment biases	Lectin/HILIC enrichment favors only subsets of glycans; under-represents low-abundance or highly sialylated/fucosylated species; co-purified non-glycopeptides reduce dynamic range.	[1,3]
4. Semi-quantitative datasets	Lack of isotopically labeled standards; matrix effects in CSF/plasma; batch effects; inter-laboratory variability limit absolute quantification.	[3,59]
5. Proteoform search-space inflation	Assigning spectra to specific glycoproteoforms increases FDR; peptide-centric pipelines struggle with co-existing PTMs (*O*-GlcNAc, phosphorylation) and site-occupancy reporting.	[59]
6. Sample-handling artifacts	Partial desialylation or de-*N*-glycosylation during preparation distorts native profiles; limited biofluid material restricts replication.	[1,3]
7. Limited spatial resolution	Bulk proteomics lacks regional and cell-type resolution needed to map glycosylation to tauopathy topography.	[1,3,59]

CSF, cerebrospinal fluid; FDR, false discovery rate, HILIC, hydrophilic interaction liquid chromatography; PTMs, post-translational modifications.

**Table 2 biomedicines-13-02884-t002:** Emerging solutions in tau tau glycoproteomics.

Solution Category	Description/Technologies	References
1. Hybrid fragmentation for improved site localization	ETD, EThcD, AI-ETD, stepped-HCD provide complementary backbone plus glycan information, improving site resolution.	[61]
2. Ion mobility for isomer separation and sensitivity	FAIMS and TIMS-PASEF separate isomeric/isobaric glycopeptides, increasing sensitivity and reducing chemical noise in CSF.	[62]
3. Orthogonal structural decoding	Exoglycosidase arrays, permethylation, linkage-specific sialic acid chemistry constrain structural assignments.	[63]
4. Targeted quantitative workflows	PRM, DIA-SWATH, and stable-isotope internal standards enable reproducible multi-center-ready quantification.	[61,62]
5. Top-down and native MS	Resolve intact proteoforms containing combined glyco- and phospho-states, clarifying PTM crosstalk on tau.	[62,63]
6. Spatial glycoproteomics	MALDI imaging, lectin-guided IMS, laser capture microdissection + LC-MS provide anatomical resolution aligned with Braak staging.	[63]
7. Advanced bioinformatics and standardization	Open/offset searches, glycan-aware FDR control, site-occupancy reporting; improved metadata standards and enzyme-toggle controls enhance transparency.	[61,62,63]

AI-ETD, activated ion electron transfer dissociation; CSF, cerebrospinal fluid; DIA-SWATH, data-independent scquisition—sequential window acquisition of all theoretical mass spectra; ETD, electron transfer dissociation; EThcD, electron-transfer and higher-energy collision dissociation, FAIMS, field asymmetric ion mobility spectrometry, FDR, false discovery rate; HCD, higher-energy collisional dissociation; IMS, imaging mass spectrometry; LC, liquid chromatography; PRM, parallel reaction monitoring, PTM, post-translational modification; TIMS-PASEF, trapped ion mobility spectrometry—parallel accumulation serial fragmentation.

**Table 3 biomedicines-13-02884-t003:** Disease-specific glycosylation signatures in tauopathies.

Disorder	Tau Isoform/Filament	Tau *N*-Glycosylation (Trend/Examples)	Tau *O*-GlcNAcylation (Trend)	Phosphorylation Pattern (Illustrative)	Predominant Regions/Phenotype	CSF/Serum Glyco-Biomarker Notes
Alzheimer’s disease (AD)	Mixed 3R/4R; PHFs	Aberrant *N*-glycosylation detected in NFTs; contributes to aggregation and primes hyperphosphorylation (e.g., Asn sites reported).	Reduced *O*-GlcNAcylation vs. controls; loss of protective competition with phosphorylation.	Broad phospho-tau engagement (e.g., S202/T205, T231, S396/404).	Medial temporal → associative neocortex; progressive cognitive decline.	CSF shows disease-linked glycoform shifts; glycoproteomics distinguishes AD from controls and supports diagnostic potential.
Progressive supranuclear palsy (PSP)	Predominantly 4R; straight filaments	Enrichment of high-mannose *N*-glycans on tau reported relative to AD, indicating distinct enzymatic dysregulation.	*O*-GlcNAc changes reported but generally less reduced than in AD (comparatively).	Phospho-site usage differs from AD; strong 4R context.	Brainstem, basal ganglia; axial rigidity, vertical gaze palsy.	Distinct glycan patterns vs. AD suggest differential-diagnosis utility when combined with phospho-tau.
Corticobasal degeneration (CBD)	Predominantly 4R; straight filaments	Relative enrichment of complex-type *N*-glycans vs. PSP; disease-specific *N*-glycan profile.	*O*-GlcNAc alterations present; comparative magnitude uncertain.	4R-biased phospho-signature; corticobasal distribution.	Asymmetric cortical and basal ganglia involvement; apraxia, dystonia.	Glyco-profiling may separate CBD from PSP/AD in research cohorts.
FTLD-tau (e.g., Pick’s disease/other subtypes)	Often 3R (Pick’s); mixed in others; diverse inclusion types	Abnormal tau glycosylation reported; pattern varies by subtype/glial vs. neuronal predominance.	*O*-GlcNAc deficits also observed in FTLD tissue, linking metabolism to tau pathology.	Subtype-specific phospho-maps; Pick bodies in 3R disease.	FTLD; behavioral/language syndromes.	Distinct glyco-signatures vs. AD/PSP/CBD are emerging in glycoproteomics studies.

3R/4R, three/four microtubule-binding repeats; AD, Alzheimer’s disease; CBD, corticobasal degeneration; CSF, cerebrospinal fluid; FTLD, frontotemporal lobar degeneration; NFT, neurofibrillary tangle; *O*-GlcNAc, *O*-linked *N*-acetylglucosamine; PHFs, paired helical filaments, PSP, progressive supranuclear palsy.

**Table 4 biomedicines-13-02884-t004:** Summary of glycosylation processes in AD and tauopathies.

Process/Target	Type of Glycosylation	Pathological Consequence	Biomarker/Therapeutic Relevance	Key References
Tau protein	*N*-glycosylation	Promotes hyperphosphorylation and aggregation	Detected in NFTs; biomarker potential	[12,82]
Tau protein	*O*-GlcNAcylation	Protective, reduces phosphorylation and aggregation	Reduced in AD brains; OGA inhibitors in trials	[20,59,68]
APP	*N*-glycosylation	Alters trafficking; increases amyloidogenic cleavage	Potential target for secretase regulation	[5,8,26]
BACE1 (β-secretase)	*N*-glycosylation	Stabilizes enzyme, promotes Aβ production	Inhibition reduces Aβ levels	[17,24,32,47,77,85]
Nicastrin (γ-secretase)	*N*-glycosylation	Modulates substrate binding and Aβ species ratio	Glycan-targeting therapies under exploration	[18,55,64,80]
Synaptic receptors (NMDA, AMPA)	*N*-glycosylation	Controls receptor trafficking and function	Aberrant glycosylation increases Aβ vulnerability	[38,61]
NCAM (adhesion)	Polysialylation	Regulates neurite outgrowth and synaptic plasticity	Reduced in AD hippocampus	[32,64,68]
Immune receptors (TREM2, CD33)	*N*-glycosylation	Controls stability and microglial response	Mutations affect AD risk	[27,59,67]
Cytokines (IL-6, TNF-α)	*N*-glycosylation	Regulates secretion and signaling	Altered profiles detected in CSF	[28,41]
Complement proteins (C1q, C3)	Sialylation	Regulates activation and synaptic pruning	Aberrant glycosylation enhances synapse loss	[9,12,54,60]
Enzymes (OGT, OGA)	Glycosylation enzymes	Balances *O*-GlcNAcylation/phosphorylation	Biomarker and therapeutic target	[83,88]

Aβ, amyloid-β; AD, Alzheimer’s disease; AMPA, α-amino-3-hydroxy-5-methyl-4-isoxazolepropionic acid receptor; APP, amyloid precursor protein; BACE1, β-site APP-cleaving enzyme; CD33, cluster of differentiation 33; CSF, cerebrospinal fluid; IL-6, interleukin-6; OGA, *O*-GlcNAcase; OGT, *O*-GlcNAc transferase; NCAM, neural cell adhesion molecule; NFT, neurofibrillary tangle; NMDA, *N*-methyl-d-aspartate receptor; TNF-α, tumor necrosis factor-alpha; TREM2, triggering receptor expressed on myeloid cells 2.

**Table 5 biomedicines-13-02884-t005:** Recommended reporting standards, disease signatures, validation rules, and translation framework for tau glycoproteomics.

Category	Item	Description	References
A. Minimum reporting set (per study)	Tau isoform and filament context	Report tau isoform composition (3R, 4R, mixed) and filament type (paired helical filaments vs. straight filaments), together with sampled brain regions.	[4,26]
	Site-resolved PTM map	Provide integrated PTM map of the same specimens: (i) *N*-glycosylation (Asn), (ii) *O*-GlcNAc (Ser/Thr), and (iii) phosphorylation at canonical AD sites (T231, S202/T205, S396/404).	[1,2,72]
	Co-occupancy or competition metrics	Quantify *O*-GlcNAc vs. phospho-tau co-occupancy or competition at adjacent residues using MS or orthogonal immunoassays (e.g., antibody–lectin hybrids).	[74,75,76,77,78,79,80,81,82,83,84,85,86]
	Specimen matrix and assay performance	Report matrix (brain region, CSF, plasma) and assay characteristics (LoD, LoQ, spike-recovery, matrix effects) to allow cross-study comparison.	[99,100]
B. Disease-specific signatures to report	Alzheimer’s disease	Mixed 3R/4R tau, PHF filaments; increased *N*-glycosylation with reduced *O*-GlcNAc; broad phospho-site engagement.	[4,104]
	PSP/CBD	Predominantly 4R tau, straight filaments; restricted *N*-glycan complexity; distinct phospho-site profile; evaluate whether *O*-GlcNAc loss is less pronounced than in AD.	[7,26,51]
	Pick’s disease and others	Specify 3R tau (e.g., Pick’s) vs. mixed isoforms; document glial vs. neuronal glycosylation patterns.	[52,53,54]
C. Assay and antibody validation rules	Phospho-tau antibodies	Validate antibodies such as AT8 (pS202/pT205), PHF-1 (pS396/pS404), CP13 (pS202) for IHC and CSF. Include epitope-blocking peptide and phosphatase-treatment controls.	[1,2,8,56]
	Glyco-tau antibodies	Because glyco-tau antibodies remain research-grade, use: (i) enzymatic toggling (PNGase F; OGA/OGT modulation), (ii) lectin co-capture, and (iii) MS confirmation in the same samples (especially for CSF/serum).	[69,70,71]
	Fluid-based assays	Prefer antibody–lectin hybrid ELISA or IP→LC-MS/MS for glyco-site certainty in fluids; interpret IHC glyco-signal cautiously unless enzymatic controls are demonstrated.	[84]
D. Decision framework for clinical translation	Tier 1—Exploratory	MS-centric site discovery in brain tissue with matched phospho- and *O*-GlcNAc-tau data.	[115]
	Tier 2—Verification	Antibody–lectin or IP-MS assays in CSF; report full analytical validation parameters.	[112]
	Tier 3—Qualification	Multi-center CSF and plasma biomarker panels combining phospho-tau and tau glyco-epitopes with APP-related and immune glyco-markers for differential diagnosis (AD vs. PSP/CBD).	[121]

AD, Alzheimer’s disease; APP, amyloid precursor protein; CBD, corticobasal degeneration; CP13, antibody recognizing phosphorylated tau at Ser202; CSF, cerebrospinal fluid; ELISA, enzyme-linked immunosorbent assay; IHC, immunohistochemistry; IP, immunoprecipitation; LC, liquid chromatography; LoD, limit of detection; LoQ, limit of quantification; PHF, paired helical filaments; PHF-1, antibody recognizing phosphorylated tau at Ser396/Ser404; PNGase, peptide: N-glycosidase F; PSP, progressive supranuclear palsy, PTM, post-translational modification.

## Data Availability

No new data were created or analyzed in this study. Data sharing is not applicable to this article.

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
