# Peer review of "Unlocking the Sugar Code: Implications and Consequences of Glycosylation in Alzheimer’s Disease and Other Tauopathies"

_biomedicines, 2025, doi:10.3390/biomedicines13122884_

Round 1
Reviewer 1 Report
Comments and Suggestions for Authors
The manuscript biomedicines-3965085 entitled Unlocking the Sugar Code: Implications and Consequences of Glycosylation in Alzheimer’s Disease and Other Tauopathies by Andrei-Cristian Bondar and coworkers, reviewed here the current evidence on the implications and consequences of glycosylation in AD and other tauopathies, integrating mechanistic, pathological, and clinical findings. They also discussed advances in glycoproteomics, the interplay between glycosylation and phosphorylation, and the translational potential of targeting glycosylation pathways for diagnosis and therapy.
The review work is updated: the authors revised properly the available work.
The writing quality is rather good and catching.
The figure is clear.
The table need ot be formatted in a more readable way.
English is very good.
The Discussion and conclusion are consistent with the exposed results.
The overall merit is: very good.
Line 182: the title should be in the following page
Line 202: the CD33 glycosylation
Line 228: the title should stay in the following page
Line 351: the notes to figure 1 should stay in the same page of the figure
Line 389, table 1: the table must to be formatted in a different way. Right now is not readable and informative
Also should stay in one page.
Line 397: The accumulated evidence underscores glycosylation as a central, though historically underappreciated, driver of Alzheimer’s disease (AD) and related tauopathies. Unlike phosphorylation, which has dominated the research landscape, glycosylation provides a multidimensional regulatory layer that influences protein stability, trafficking, aggregation, and clearance [1,19,89].
Reformulate as: Growing evidences demonstrate the role of glycosylation as a central, driver of Alzheimer’s disease (AD) and related tauopathies. Unlike phosphorylation, glycosylation provides a multidimensional regulatory layer that influences protein stability, trafficking, aggregation, and clearance [1,19,89].
Line 402: the title should stay with the dfollowing text
Line 426: Neuroinflammation is increasingly viewed not as a secondary phenomenon but as a core contributor to neurodegeneration
Rewrite as: Neuroinflammation play a key action in the neurodegenerative process.
Line 495: Systems biology approaches: Glycosylation does not act in isolation
Systems biology approaches: Glycosylation does not act isolated.
Author Response
We sincerely thank you for your insightful comments and suggestions.
The revisions made in response have substantially improved the quality of our manuscript.

Reviewer 2 Report
Comments and Suggestions for Authors
Tau protein is a microtubule-associated protein that helps maintain the stability of microtubules, which are essential components of neurons for preserving their shape and facilitating nutrient transport. This review manuscript presents a comprehensive overview of the emerging role of glycosylation in Alzheimer’s disease (AD) and related tauopathies. It also effectively integrates molecular, pathological, and translational perspectives, emphasizing the importance of post-translational modifications in neurodegenerative mechanisms. Overall, the review provides a rationale for exploring glycosylation as both a mechanistic and therapeutic target in tauopathies. However, you may wish to further emphasize the novelty or knowledge gap that your review addresses—specifically, what aspect of glycosylation research advances beyond or complements phosphorylation studies. Additionally, please consider addressing the following points:
- Make more information to clarify and compare the differences between Alzheimer’s disease (AD) and other tauopathies in terms of Tau protein pathology.
- Since there are numerous post-translational modification sites on Tau protein, you could discuss the availability and reliability of antibodies targeting Tau glycosylation and phosphorylation sites. Are such antibodies validated or suitable for detection in serum, cerebrospinal fluid (CSF), or immunohistochemistry (IHC) applications?
- As shown in Figure 1 and Table 1, only N-glycosylation appears to promote Tau hyperphosphorylation and aggregation. It would be better to compare the differences between Tau protein glycosylation and phosphorylation sites, and to discuss how these modifications contribute differently to the mechanisms of Alzheimer’s disease (AD) and related tauopathies. Overall, Figure 1 is too simple to provide new information — the correlations between Tau N- or O-glycosylation and hyperphosphorylation, as well as the specific sites involved, need more detailed discussion.
- You could also set up a Guidelines for tauopathies to emphasize the novelty or gap your review addresses—what specific aspect of glycosylation research advances beyond existing reviews, in order to reach a consensus for tau pathology (If possible, you may provide more Figure or Table).
Author Response
Comment 1: Make more information to clarify and compare the differences between Alzheimer’s disease (AD) and other tauopathies in terms of Tau protein pathology.
Response 1: Thank you for your suggestion, we have inserted a paragraph at the end of the 2.3 subsection to provide more information regarding the differences to increase clarity.
Comment 2: Since there are numerous post-translational modification sites on Tau protein, you could discuss the availability and reliability of antibodies targeting Tau glycosylation and phosphorylation sites. Are such antibodies validated or suitable for detection in serum, cerebrospinal fluid (CSF), or immunohistochemistry (IHC) applications?
Response 2: Thank you for your suggestion. We have inserted 2 paragraphs with the solicited information at the end of subsection 5.3.
Comment 3: As shown in Figure 1 and Table 1, only N-glycosylation appears to promote Tau hyperphosphorylation and aggregation. It would be better to compare the differences between Tau protein glycosylation and phosphorylation sites, and to discuss how these modifications contribute differently to the mechanisms of Alzheimer’s disease (AD) and related tauopathies. Overall, Figure 1 is too simple to provide new information — the correlations between Tau N- or O-glycosylation and hyperphosphorylation, as well as the specific sites involved, need more detailed discussion.
Response 3: Thank you for your suggestions. We have created an entire subsection to be able to provide the necessary information. We consider this helped increase the scientific value of the manuscript.
Comment 4: You could also set up a Guidelines for tauopathies to emphasize the novelty or gap your review addresses—what specific aspect of glycosylation research advances beyond existing reviews, in order to reach a consensus for tau pathology (If possible, you may provide more Figure or Table)
Response 4: We thank you for your suggestions. We have added the section 7.8 with the requested guideline proposal and Figure 2 with site-resolved crosstalk map on tau.

Reviewer 3 Report
Comments and Suggestions for Authors
Major points:
1) This review manuscript suffers from a lack of sufficient number of figures. Please sketch at least three new figures that would neatly describe the underpinning concepts and principles. Namely, it will be interesting to illustrate the enzymatic "tug-of-war" mentioned in "This enzymatic tug-of-war suggests that targeting glycosylation enzymes may indirectly modulate tau phosphorylation, providing a dual mechanism for therapeutic intervention" (line 372).
2) The illustration depicted in Figure 1 is rather small. Please increase its size without compromising digital resolution.
Minor points:
1) Please remove bold formatting from "glycosylation" (lines 21, 49).
2) Please format "N" in "N-linked" using italics (lines 23, 84).
3) Please format "N" in "N-glycosylation" using italics (lines 23, 34, 62, 88, 94, 116, 119, 133, 135, 138, 162, 185, 198, 338, 404, 428, 460, 521).
4) Please remove bold formatting from "N-linked glycosylation (N-glycosylation)" (line 23).
5) Please format "O" in "O-linked" using italics (lines 23, 93).
6) Please remove bold formatting from "O-linked glycosylation (O-glycosylation)" (page 23).
7) Please format "O" in "O-glycosylation" using italics (lines 24, 34, 52, 138, 140, 141, 173, 340, 463)
8) Please remove bold formatting from "implications and consequences of glycosylation in AD and other tauopathies" (line 30).
9) Please change "dementia, affecting" to "dementia affecting" (line 39).
10) Please remove bold formatting from "amyloid-β (Aβ) plaques" (line 43).
11) Please remove bold formatting from "neurofibrillary tangles (NFTs)" (line 43).
12) Please remove bold formatting from "post-translational modification (PTM)" (line 47).
13) Please replace "are: N-glycosylation" with "are N-glycosylation" (line 51).
14) Please remove bold formatting from "N-glycosylation" (line 51).
15) Please remove bold formatting from "O-glycosylation" (lines 52, 138).
16) Please format "O" in "O-GlcNAcylation" using italics (lines 54, 93, 94, 95, 97, 99, 109, 116, 118, 264, 330, 332, 339, 358, 359, 362, 368, 371, 384, 410, 412, 420, 443, 461, 522).
17) Please format "N" in "N-acetylglucosamine" using italics (lines 54, 93).
18) Please change "influences" to "influences protein" (line 56).
19) Please remove bold formatting from "frontotemporal lobar degeneration (FTLD), progressive supranuclear palsy (PSP)," (page 58).
20) Please remove bold formatting from "corticobasal degeneration (CBD)" (line 59).
21) Please replace "the amyloid" with "amyloid" (line 61).
22) Please remove bold formatting from "glycoproteomics" (line 65).
23) Please remove bold formatting from "glycosyltransferases" (line 68).
24) Please remove bold formatting from "glycosidases" (line 68).
25) Please change "targets, while" to "targets while" (line 68).
26) Please remove bold formatting from "implications and consequences 71
of glycosylation in AD and other tauopathies" (line 71).
27) Please replace "Glycosylation in Alzheimer’s and Related Tauopathies" with "glycosylation in Alzheimer’s disease and related tauopathies" (line 75).
28) Please replace "Alzheimer’s disease (AD)" with "AD" (lines 77, 124, 179, 243, 310, 398).
29) Please change "post-translational modifications" to "PTMs" (lines 78).
30) Please remove bold formatting from "glycosylation is a critical determinant of tau structure, aggregation, and toxicity" (line 80).
31) Please format "N" in "N-glycosylated" using italics (lines 83, 86, 114, 128).
32) Please replace "frontotemporal lobar degeneration (FTLD), progressive supranuclear palsy (PSP), and corticobasal degeneration (CBD)" with "FTLD, PSP, and CBD" (line 103).
33) Please change "although" to something like "despite" (line 105).
34) Please format "N" in "N-glycans" using italics (lines 107, 128, 130, 143, 188, 234, 255, 315, 316).
35) Please replace "imbalance:" with "imbalance," (line 116).
36) Please format "O" in "O-GlcNAcase" using italics (lines 120, 394).
37) Please format "O" in "O-GlcNAc" using italics (lines 121, 394).
38) Please change "Amyloid-β Pathology" to "amyloid-β pathology" (line 122).
39) Please provide reference for "The amyloid cascade hypothesis proposes that abnormal processing of amyloid precursor protein (APP) initiates a series of pathogenic events culminating in Alzheimer’s disease (AD)" (line 123).
40) Please replace "amyloid-β (Aβ)" with "Aβ" (lines 126, 341).
41) Please change "APP" to "Amyloid Precursor Protein" (line 127).
42) Please replace "β-secretase - β-site" with something like "the β-secretase enzyme, β-site" (line 131).
43) Please format "O" in "O-GalNAc" using italics (line 139).
44) Please change "glycosylated; glycan structures" to something like "glycosylated. The glycan structures of nicastrin" (line 146).
45) Please replace "Aβ" with "amyloid-β" (line 149).
46) Please remove bold formatting from "receptor-mediated endocytosis" (line 155).
47) Please change "its" to "their" (line 157).
48) Please replace "NMDA" with "N-methyl-D-aspartate (NMDA)" (line 162) and "N-methyl-D-aspartate (NMDA)" with "NMDA" (line 184).
49) Please remove bold formatting from "NMDA receptors" (line 162).
50) Please replace "expression; loss" with "expression as loss" (line 163).
51) Please change "[20].Similarly" to "[20]. Similarly" (line 164).
52) Please replace "TREM2" with "triggering receptor expressed on myeloid cells 2 (TREM2)" (line 167) and "Triggering receptor expressed on myeloid cells 2 (TREM2)" with "TREM2" (line 197).
53) Please define abbreviation for "TREM2" (line 167), "CD33" (line 167), "AMPA" (line 188), "GFAP" in "GFAP-interacting" (line 206).
54) Please remove bold formatting from "targeting glycosylation pathways" (line 171).
55) Please specify the "deposition" mentioned in "Additionally, blocking RAGE glycosylation has been proposed to limit Aβ
transport into the brain and reduce deposition" (line 175). Deposition of what?
56) Please replace "Synaptic Function, and Neuroinflammation" with "synaptic function, and neuroinflammation" (line 177).
57) Please format "N" in "N-methyl-D-aspartate" using italics (lines 184, 391).
58) Please format "D" in "N-methyl-D-aspartate" using small caps (lines 184, 391).
59) Please change "amyloid-β (Aβ)-induced" to "Aβ-induced" (line 187).
60) Please replace "properties; altered" with "properties. altered" (line 188).
61) Please provide reference for "Another example is CD33, a sialic acid-binding receptor implicated in AD genetic risk" (line 201).
62) Please change "signaling; aberrant" to something like "signaling while aberrant" (line 202).
63) Please provide reference for "C1q, the initiator of the classical pathway, carries sialylated glycans that regulate binding to immune complexes and synaptic elements" (line 221).
64) Please replace "RAGE (receptor for advanced glycation end products)" with "RAGE" (line 230).
65) Please remove bold formatting from "[3]." (line 233).
66) Please change "Glycoproteomics and Biomarker Discovery" to "glycoproteomics and biomarker discovery" (line 242).
67) Please replace "cerebrospinal fluid (CSF)" with "CSF" (line 244).
68) Please remove bold formatting from "[32]." (line 246).
69) Please change "amyloid precursor protein (APP)" to "APP" (lines 256, 312).
70) Please remove bold formatting from "O-GlcNAcylation status of tau" (line 264).
71) Please replace "frontotemporal lobar degeneration (FTLD) and progressive supranuclear palsy (PSP)" with "FTLD and PSP" (line 267).
72) Please remove bold formatting from "frontotemporal lobar degeneration (FTLD)" (line 267).
73) Please remove bold formatting from "progressive supranuclear palsy (PSP)" (line 268).
74) Please change "mass spectrometry (MS)" to "MS" (line 271).
75) Please provide reference for "Methods such as electron-transfer dissociation (ETD) and higher-energy collisional dissociation (HCD) provide site-specific glycan characterization with high sensitivity" (line 272).
76) Please remove bold formatting from "IgG Fc glycosylation" (line 283).
77) Please replace "challenges: glycan" with "challenges. Glycan" (line 298).
78) Please change "Regulators of Glycosylation in Neurodegeneration" to "regulators of glycosylation in neurodegeneration" (line 307).
79) Please replace "AD" with "Alzheimer’s disease" (line 314).
80) Please format "N" in "N-acetylglucosaminyltransferases" using italics (line 315).
81) Please change "progressive supranuclear palsy (PSP) and corticobasal degeneration (CBD)" to "PSP and CBD" (line 323).
82) Please format "N" in "β-N-acetylglucosaminidase" using italics (line 330).
83) Please replace "facilitating" with "facilitating its" (line 331).
84) Please change "plasticity; their" to "plasticity. Their" (line 343).
85) Please replace "RAGE" with "and RAGE" (lines 344, 468).
86) Please change "hexosamine biosynthetic pathway (HBP)" to "HBP" (lines 346, 357).
87) The two lines representing the arrow leading from "Tau N-glycosylation" seem to be disconnected at the corner in Figure 1. Please fix.
88) Similarly, the two lines representing the arrow leading from "Tau O-GlcNAcylation" seem to be disonnected at the corner in Figure 1. Please fix.
89) Please replace "APP Glycosylation" with "APP glycosylation", "RAGE Glycosylation" with "RAGE glycosylation", "Beta Amyloid Fibres" with "Amyloid beta fibrils", "Amyloid Beta Plaque" with "Amyloid beta plaque" in Figure 1.
90) Please provide abbreviation for "HBP", "OGT", "OGA", "GnTs" in the legend to Figure 1.
91) From "Schematic Overview of Glycosylation in Alzheimer’s Disease and Tauopathies, illustrating how glycosylation affects tau, APP/Aβ, synaptic and immune receptors, and the enzymatic/metabolic regulators (HBP, OGT, OGA, GnTs)" (line 351) is not unequivocally clear whether the authors mean to say "enzymatic and metabolic regulators" or "enzymatic or metabolic regulators"? Please fix.
92) Please change "Overview of Glycosylation in Alzheimer’s Disease and Tauopathies" to "overview of glycosylation in Alzheimer’s disease and tauopathies" (line 351).
93) Please replace "synaptic" with "synaptic," (line 352).
94) Please change "GnTs" to "and GnTs" (line 353).
95) Please remove bold formatting from "Created with bioRender." (line 353).
96) Please replace "Created with bioRender. Acronyms in figure defined: APP (amyloid precursor protein), RAGE (receptor for advanced glycation end products)" with "APP, amyloid precursor protein; RAGE, receptor for advanced glycation end products. Created with BioRender" (line 353).
97) Please format "N" in "UDP-N-acetylglucosamine" using italics (line 357).
98) Please change "glycogen synthase kinase-3β (GSK3β)" to "GSK3β" (line 369).
99) Please replace "Sialidase inhibitors, by preserving polysialylation, may" with "By preserving polysialylation, sialidase inhibitors may" (line 384).
100) Please replace "Alzheimer’s Disease" with "AD" (lines 387, 508).
101) Please change "Glycosylation Processes in Alzheimer’s Disease and Tauopathies" to "glycosylation processes in Alzheimer’s disease and tauopathies" (line 389).
102) Please replace "Acronyms in table defined: APP" with "APP" (line 390).
103) Please sort abbreviations according to the alphabetical order in Table 1.
104) Please change "APP (amyloid precursor protein)," to "APP, amyloid precursor protein;" (line 390).
105) Please replace "Aβ (amyloid-β)," with "Aβ, amyloid-β;" (line 390).
106) Please change "NFT (neurofibrillary tangle)," to "NFT, neurofibrillary tangle;" (line 390).
107) Please replace "NMDA (N-methyl-D-aspartate receptor)," with "NMDA, N-methyl-D-aspartate receptor;" (line 391).
108) Please change "AMPA (α-amino-3-hydroxy-5-methyl-4-isoxazolepropionic acid receptor)," to "AMPA, α-amino-3-hydroxy-5-methyl-4-isoxazolepropionic acid receptor;" (line 391).
109) Please replace "NCAM (neural cell adhesion molecule)," with "NCAM, neural cell adhesion molecule;" (line 392).
110) Please change "TREM2 (triggering receptor expressed on myeloid cells 2)," to "TREM2, triggering receptor expressed on myeloid cells 2;" (line 392).
111) Please replace "CD33 (cluster of differentiation 33)," with "CD33, cluster of differentiation 33;" (line 393).
112) Please change "IL-6 (interleukin-6)," to "IL-6, interleukin-6;" (line 393).
113) Please replace "TNF-α (tumor necrosis factor-alpha)," with "TNF-α, tumor necrosis factor-alpha;" (line 393).
114) Please change "CSF (cerebrospinal fluid)," to "CSF, cerebrospinal fluid;" (line 394).
115) Please replace "OGT (O-GlcNAc transferase)," with "OGT, O-GlcNAc transferase;" (line 394).
116) Please change "OGA (O-GlcNAcase)" to "OGA, O-GlcNAcase" (line 394).
117) Please replace "“common denominator.”" with "“common denominator”." (line 408).
118) Please change "it" to "which" (line 411).
119) Please replace "strategies, interventions" with something like "strategies. Hence, interventions" (line 415).
120) Please change "hexosamine biosynthetic pathway" to "HBP" (lines 420, 471).
121) Please replace "stability; mutations" with "stability. Mutations" (line 428).
122) Please change "remain: glycan" to "remain. Glycan" (line 439).
123) Please replace "several" with "the following" (line 458).
124) Please change "emerge:" to "emerge" (line 459).
125) Please replace "Tau regulation: Aberrant" with "Aberrant" (line 460).
126) Please change "aggregation, while" to "aggregation while" (line 461).
127) Please replace "Amyloid precursor protein (APP) and amyloid-β (Aβ): N-" with "N-" (line 463).
128) Please format "N" in "N-" using italics (line 463).
129) Please change "Synaptic and immune regulation: Glycosylation" to "Glycosylation" (line 466).
130) Please replace "Enzymatic control and metabolism: Enzymes" with "Enzymes" (line 469).
131) Please change "Biomarker potential: Glycoproteomic" to "Glycoproteomic" (line 473).
132) Please replace "Directions" with "directions" (line 479).
133) Please change "remain:" to "remain." (line 481).
134) Please replace "Clarifying temporal dynamics: Longitudinal" with "Longitudinal" (line 482).
135) Please change "Expanding glycoproteomic profiling: High-resolution" to "High-resolution" (line 486).
136) Please replace "Targeted therapeutics: Clinical" with "Clinical" (line 490).
137) Please change "cautiously, with" to "cautiously with" (line 491).
138) Please replace "glycosylation (protective vs. pathogenic)" with "glycosylation, protective vs. pathogenic" (line 492).
139) Please change "Systems biology approaches: Glycosylation" to "Glycosylation" (line 495).
140) Please replace "Expanding to non-AD tauopathies: Comparative" with "Comparative" (line 499).
141) Please change "Bridging bench to bedside: Translation" to "Translation" (line 503).
142) Please replace "processing, while" with "processing while" (line 521).
143) Please change "cerebrospinal fluid" to "CSF" (line 527).
144) Please replace "M.C." with "M.C.," (lines 536, 537).
145) Please change "M.B., T.C." to "M.B., and T.C." (lines 537, 538, 539 2x, 540, 541, 542).
146) Please replace "M.B. and" with "M.B., and" (line 537).
147) Please change "investigation, A.C.B., M.P.I., M.C., A.B." to "investigation, A.C.B., M.P.I., M.C., and A.B." (line 538).
148) Please replace "A.C.B., M.P.I." with "A.C.B. and M.P.I." (line 539).
149) Please change "editing, M.C., A.B." to "editing, M.C. and A.B." (line 540).
150) Please replace "M.P.I., M.C., A.B." with "M.P.I., M.C., and A.B." (line 540).
Author Response
MAJOR POINTS
Comment 1: This review manuscript suffers from a lack of sufficient number of figures. Please sketch at least three new figures that would neatly describe the underpinning concepts and principles. Namely, it will be interesting to illustrate the enzymatic "tug-of-war" mentioned in "This enzymatic tug-of-war suggests that targeting glycosylation enzymes may indirectly modulate tau phosphorylation, providing a dual mechanism for therapeutic intervention" (line 372).
Response 1: Thank you for your suggestions. We inserted new figures and a new table to make the presented information more accessible. Figure 2 addresses the aforementioned "tug-of-war" mentioned by our reviewer.
Comment 2: The illustration depicted in Figure 1 is rather small. Please increase its size without compromising digital resolution.
Response 2: Thank you for your suggestion. We have increased the size of the figure without compromising quality.
MINOR POINTS
Comment 1: Please remove bold formatting from "glycosylation" (lines 21, 49).
Response 1: Thank you for your suggestion. We have removed the formatting from "glycosylation" (lines 21, 49).
Comment 2: Please format "N" in "N-linked" using italics (lines 23, 84).
Response 2: Thank you for your suggestion. We have formatted "N" in "N-linked" using italics.
Comment 3: Please format "N" in "N-glycosylation" using italics (lines 23, 34, 62, 88, 94, 116, 119, 133, 135, 138, 162, 185, 198, 338, 404, 428, 460, 521).
Response 3: Thank you for your suggestion. We have made all the necessary adjustments.
Comment 4: Please remove bold formatting from "N-linked glycosylation (N-glycosylation)" (line 23).
Response 4: Thank you for your suggestion. We have removed the bold formatting.
Comment 5: Please format "O" in "O-linked" using italics (lines 23, 93).
Response 5: Thank yourfor your suggestion. We formatted the "O" in "O-linked" using italics.
Comment 6: Please remove bold formatting from "O-linked glycosylation (O-glycosylation)" (page 23).
Response 6: Thank you for your suggestion. We removed the bold formatting from "O-linked glycosylation (O-glycosylation)" (line 23).
Comment 7: Please format "O" in "O-glycosylation" using italics (lines 24, 34, 52, 138, 140, 141, 173, 340, 463).
Response 7: Thank you for your suggestion. We have formatted the "O" in "O-glycosylation" using italics.
Comment 8: Please remove bold formatting from "implications and consequences of glycosylation in AD and other tauopathies" (line 30).
Response 8: Thank you for your suggestion. We removed the bold formatting from "implications and consequences of glycosylation in AD and other tauopathies".
Comment 9: Please change "dementia, affecting" to "dementia affecting" (line 39).
Response 9: Thank you for your suggestion. We replaced "dementia, affecting" with "dementia affecting".
Comment 10: Please remove bold formatting from "amyloid-β (Aβ) plaques" (line 43).
Response 10: Thank you for your suggestion. We removed the bold formatting from "amyloid-β (Aβ) plaques".
Comment 11: Please remove bold formatting from "neurofibrillary tangles (NFTs)" (line 43).
Response 11: Thank you for your suggestion. We removed the bold formatting from "neurofibrillary tangles (NFTs)".
Comment 12: Please remove bold formatting from "post-translational modification (PTM)" (line 47).
Response 12: Thank you for your suggestion. We removed the bold formatting from "post-translational modification (PTM)".
Comment 13: Please replace "are: N-glycosylation" with "are N-glycosylation" (line 51).
Response 13: Thank you for your suggestion. We replaced "are: N-glycosylation" with "are N-glycosylation".
Comment 14: Please remove bold formatting from "N-glycosylation" (line 51).
Response 14: Thank you for your suggestion. We removed the bold formatting from "N-glycosylation".
Comment 15: Please remove bold formatting from "O-glycosylation" (lines 52, 138).
Response 15: Thank you for your suggestion. We removed the bold formatting from "O-glycosylation".
Comment 16: Please format "O" in "O-GlcNAcylation" using italics (lines 54, 93, 94, 95, 97, 99, 109, 116, 118, 264, 330, 332, 339, 358, 359, 362, 368, 371, 384, 410, 412, 420, 443, 461, 522).
Response 16: Thank you for your suggestion. We formatted the "O" in "O-GlcNAcylation" using italics.
Comment 17: Please format "N" in "N-acetylglucosamine" using italics (lines 54, 93).
Response 17: Thank you for your suggestion. We formatted the "N" in "N-acetylglucosamine" using italics.
Comment 18: Please change "influences" to "influences protein" (line 56).
Response 18: Thank you for your suggestion. We changed "influences" to "influences protein folding...".
Comment 19: Please remove bold formatting from "frontotemporal lobar degeneration (FTLD), progressive supranuclear palsy (PSP)," (page 58).
Response 19: Thank you for your suggestion. We removed the bold formatting from "frontotemporal lobar degeneration (FTLD), progressive supranuclear palsy (PSP)," (line 58).
Comment 20: Please remove bold formatting from "corticobasal degeneration (CBD)" (line 59).
Response 20: Thank you for your suggestion. We removed the bold formatting from "corticobasal degeneration (CBD)".
Comment 21: Please replace "the amyloid" with "amyloid" (line 61).
Response 21: Thank you for your suggestion. We replaced "the amyloid" with "amyloid".
Comment 22: Please remove bold formatting from "glycoproteomics" (line 65).
Response 22: Thank you for your suggestion. We removed the bold formatting from "glycoproteomics".
Comment 23: Please remove bold formatting from "glycosyltransferases" (line 68).
Response 23: Thank you for your suggestion. We removed the bold formatting from "glycosyltransferases".
Comment 24: Please remove bold formatting from "glycosidases" (line 68).
Response 24: Thank you for your suggestion. We removed the bold formatting from "glycosidases".
Comment 25: Please change "targets, while" to "targets while" (line 68).
Response 25: Thank you for your suggestion. We changed "targets, while" to "targets while".
Comment 26: Please remove bold formatting from "implications and consequences 71
of glycosylation in AD and other tauopathies" (line 71).
Response 26: Thank you for your suggestion. We removed the bold formatting from "implications and consequences of glycosylation in AD and other tauopathies".
Comment 27: Please replace "Glycosylation in Alzheimer’s and Related Tauopathies" with "glycosylation in Alzheimer’s disease and related tauopathies" (line 75).
Response 27: Thank you for your suggestion. We replaced "Glycosylation in Alzheimer’s and Related Tauopathies" with "glycosylation in Alzheimer’s disease and related tauopathies".
Comment 28: Please replace "Alzheimer’s disease (AD)" with "AD" (lines 77, 124, 179, 243, 310, 398).
Response 28: Thank you for your suggestion. We replaced "Alzheimer’s disease (AD)" with "AD".
Comment 29: Please change "post-translational modifications" to "PTMs" (lines 78).
Response 29: Thank you for your suggestion. We change "post-translational modifications" to "PTMs".
Comment 30: Please remove bold formatting from "glycosylation is a critical determinant of tau structure, aggregation, and toxicity" (line 80).
Response 30: Thank you for your suggestion. We removed the bold formatting from "glycosylation is a critical determinant of tau structure, aggregation, and toxicity".
Comment 31: Please format "N" in "N-glycosylated" using italics (lines 83, 86, 114, 128).
Response 31: Thank you for your suggestion. We formatted the "N" in "N-glycosylated" using italics.
Comment 32: Please replace "frontotemporal lobar degeneration (FTLD), progressive supranuclear palsy (PSP), and corticobasal degeneration (CBD)" with "FTLD, PSP, and CBD" (line 103).
Response 32: Thank you for your suggestion. We replaced "frontotemporal lobar degeneration (FTLD), progressive supranuclear palsy (PSP), and corticobasal degeneration (CBD)" with "FTLD, PSP, and CBD".
Comment 33: Please change "although" to something like "despite" (line 105).
Response 33: Thank you for your suggestion. We changed "although" with "despite".
Comment 34: Please format "N" in "N-glycans" using italics (lines 107, 128, 130, 143, 188, 234, 255, 315, 316).
Response 34: Thank you for your suggestion. We formatted the "N" in "N-glycans" using italics.
Comment 35: Please replace "imbalance:" with "imbalance," (line 116).
Response 35: Thank you for your suggestion. We replaced "imbalance:" with "imbalance,".
Comment 36: Please format "O" in "O-GlcNAcase" using italics (lines 120, 394).
Response 36: Thank you for your suggestion. We formatted the "O" in "O-GlcNAcase" using italics.
Comment 37: Please format "O" in "O-GlcNAc" using italics (lines 121, 394).
Response 37: Thank you for your suggestion. We formatted the "O" in "O-GlcNAc" using italics.
Comment 38: Please change "Amyloid-β Pathology" to "amyloid-β pathology" (line 122).
Response 38: Thank you for your suggestion. We changed "Amyloid-β Pathology" to "amyloid-β pathology".
Comment 39: Please provide reference for "The amyloid cascade hypothesis proposes that abnormal processing of amyloid precursor protein (APP) initiates a series of pathogenic events culminating in Alzheimer’s disease (AD)" (line 123).
Response 39: Thank you for your suggestion. We provided reference [8] for "The amyloid cascade hypothesis proposes that abnormal processing of amyloid precursor protein (APP) initiates a series of pathogenic events culminating in Alzheimer’s disease (AD)".
Comment 40: Please replace "amyloid-β (Aβ)" with "Aβ" (lines 126, 341).
Response 40: Thank you for your suggestion. We replaced "amyloid-β (Aβ)" with "Aβ".
Comment 41: Please change "APP" to "Amyloid Precursor Protein" (line 127).
Response 41: Thank you for your suggestion. We changed "APP" to "Amyloid Precursor Protein".
Comment 42: Please replace "β-secretase - β-site" with something like "the β-secretase enzyme, β-site" (line 131).
Response 42: Thank you for your suggestion. We replaced "β-secretase - β-site" with "β-site β-secretase enzyme".
Comment 43: Please format "O" in "O-GalNAc" using italics (line 139).
Response 43: Thank you for your suggestion. We formatted the "O" in "O-GalNAc" using italics.
Comment 44: Please change "glycosylated; glycan structures" to something like "glycosylated. The glycan structures of nicastrin" (line 146).
Response 44: Thank you for your suggestion. We changed "glycosylated; glycan structures" to "glycosylated. The glycan structures".
Comment 45: Please replace "Aβ" with "amyloid-β" (line 149).
Response 45: Thank you for your suggestion. We replaced "Aβ" with "amyloid-β".
Comment 46: Please remove bold formatting from "receptor-mediated endocytosis" (line 155).
Response 46: Thank you for your suggestion. We removed the bold formatting from "receptor-mediated endocytosis".
Comment 47: Please change "its" to "their" (line 157).
Response 47: Thank you for your suggestion. We replaced "its" with "their".
Comment 48: Please replace "NMDA" with "N-methyl-D-aspartate (NMDA)" (line 162) and "N-methyl-D-aspartate (NMDA)" with "NMDA" (line 184).
Response 48: Thank you for your suggestion. We performed both replacements accordingly.
Comment 49: Please remove bold formatting from "NMDA receptors" (line 162).
Response 49: Thank you for your suggestion. We removed the bold formatting from "NMDA receptors".
Comment 50: Please replace "expression; loss" with "expression as loss" (line 163).
Response 50: Thank you for your suggestion. We replaced "expression; loss" with "expression as loss".
Comment 51: Please change "[20].Similarly" to "[20]. Similarly" (line 164).
Response 51: Thank you for your suggestion. We added the requested space between "[20]." and "Similarly".
Comment 52: Please replace "TREM2" with "triggering receptor expressed on myeloid cells 2 (TREM2)" (line 167) and "Triggering receptor expressed on myeloid cells 2 (TREM2)" with "TREM2" (line 197).
Response 52: Thank you for your suggestion. We performed both replacements accordingly.
Comment 53: Please define abbreviation for "TREM2" (line 167), "CD33" (line 167), "AMPA" (line 188), "GFAP" in "GFAP-interacting" (line 206)
Response 53: Thank you for your suggestion. We defined all the abbreviations accordingly.
Comment 54: Please remove bold formatting from "targeting glycosylation pathways" (line 171).
Response 54: Thank you for your suggestion. We removed the bold formatting from "targeting glycosylation pathways".
Comment 55: Please specify the "deposition" mentioned in "Additionally, blocking RAGE glycosylation has been proposed to limit Aβ transport into the brain and reduce deposition" (line 175). Deposition of what?
Response 55: Thank you for your suggestion. We inserted the explanation: "and reduce amyloid-β deposition within the cerebral parenchyma and vasculature."
Comment 56: Please replace "Synaptic Function, and Neuroinflammation" with "synaptic function, and neuroinflammation" (line 177).
Response 56: Thank you for your suggestion. We replaced "Synaptic Function, and Neuroinflammation" with "synaptic function, and neuroinflammation".
Comment 57: Please format "N" in "N-methyl-D-aspartate" using italics (lines 184, 391).
Response 57: Thank you for your suggestion. We formatted the "N" in "N-methyl-D-aspartate" using italics.
Comment 58: Please format "D" in "N-methyl-D-aspartate" using small caps (lines 184, 391).
Response 58: Thank you for your suggestion. We formatted the "D" in "N-methyl-D-aspartate" using small caps.
Comment 59: Please change "amyloid-β (Aβ)-induced" to "Aβ-induced" (line 187).
Response 59: Thank you for your suggestion. We changed "amyloid-β (Aβ)-induced" to "Aβ-induced".
Comment 60: Please replace "properties; altered" with "properties. altered" (line 188).
Response 60: Thank you for your suggestion. We replaced "properties; altered" with "properties. Altered".
Comment 61: Please provide reference for "Another example is CD33, a sialic acid-binding receptor implicated in AD genetic risk" (line 201).
Response 61: Thank you for your suggestion. For "Another example is CD33, a sialic acid-binding receptor implicated in AD genetic risk", we provided references: [6,8,19].
Comment 62: Please change "signaling; aberrant" to something like "signaling while aberrant" (line 202).
Response 62: Thank you for your suggestion. We changed "signaling; aberrant" to "signaling while aberrant".
Comment 63: Please provide reference for "C1q, the initiator of the classical pathway, carries sialylated glycans that regulate binding to immune complexes and synaptic elements" (line 221).
Response 63: Thank you for your suggestion. For "C1q, the initiator of the classical pathway, carries sialylated glycans that regulate binding to immune complexes and synaptic elements", we inserted references [7,11].
Comment 64: Please replace "RAGE (receptor for advanced glycation end products)" with "RAGE" (line 230).
Response 64: Thank you for your suggestion. We replaced "RAGE (receptor for advanced glycation end products)" with "RAGE".
Comment 65: Please remove bold formatting from "[3]." (line 233).
Response 65: Thank you for your suggestion. We removed the bold formatting from "[3]."
Comment 66: Please change "Glycoproteomics and Biomarker Discovery" to "glycoproteomics and biomarker discovery" (line 242).
Response 66: Thank you for your suggestion. We changed "Glycoproteomics and Biomarker Discovery" to "glycoproteomics and biomarker discovery".
Comment 67: Please replace "cerebrospinal fluid (CSF)" with "CSF" (line 244).
Response 67: Thank you for your suggestion. We replaced "cerebrospinal fluid (CSF)" with "CSF".
Comment 68: Please remove bold formatting from "[32]." (line 246).
Response 68: Thank you for your suggestion. We removed the bold formatting from "[32]."
Comment 69: Please change "amyloid precursor protein (APP)" to "APP" (lines 256, 312).
Response 69: Thank you for your suggestion. We changed "amyloid precursor protein (APP)" to "APP".
Comment 70: Please remove bold formatting from "O-GlcNAcylation status of tau" (line 264).
Response 70: Thank you for your suggestion. We removed the bold formatting from "O-GlcNAcylation status of tau".
Comment 71: Please replace "frontotemporal lobar degeneration (FTLD) and progressive supranuclear palsy (PSP)" with "FTLD and PSP" (line 267).
Response 71: Thank you for your suggestion. We replaced "frontotemporal lobar degeneration (FTLD) and progressive supranuclear palsy (PSP)" with "FTLD and PSP".
Comment 72: Please remove bold formatting from "frontotemporal lobar degeneration (FTLD)" (line 267).
Response 72: Thank you for your suggestion. We removed the bold formatting from "frontotemporal lobar degeneration (FTLD)".
Comment 73: Please remove bold formatting from "progressive supranuclear palsy (PSP)" (line 268).
Response 73: Thank you for your suggestion. We removed the bold formatting from "progressive supranuclear palsy (PSP)".
Comment 74: Please change "mass spectrometry (MS)" to "MS" (line 271).
Response 74: Thank you for your suggestion. We changed "mass spectrometry (MS)" to "MS".
Comment 75: Please provide reference for "Methods such as electron-transfer dissociation (ETD) and higher-energy collisional dissociation (HCD) provide site-specific glycan characterization with high sensitivity" (line 272).
Response 75: Thank you for your suggestion. For "Methods such as electron-transfer dissociation (ETD) and higher-energy collisional dissociation (HCD) provide site-specific glycan characterization with high sensitivity", we have provide reference [42].
Comment 76: Please remove bold formatting from "IgG Fc glycosylation" (line 283).
Response 76: Thank you for your suggestion. We removed the bold formatting from "IgG Fc glycosylation".
Comment 77: Please replace "challenges: glycan" with "challenges. Glycan" (line 298).
Response 77: Thank you for your suggestion. We replaced "challenges: glycan" with "challenges. Glycan".
Comment 78: Please change "Regulators of Glycosylation in Neurodegeneration" to "regulators of glycosylation in neurodegeneration" (line 307).
Response 78: Thank you for your suggestion. We changed "Regulators of Glycosylation in Neurodegeneration" to "regulators of glycosylation in neurodegeneration".
Comment 79: Please replace "AD" with "Alzheimer’s disease" (line 314).
Response 79: Thank you for your suggestion. We replaced "AD" with "Alzheimer’s disease".
Comment 80: Please format "N" in "N-acetylglucosaminyltransferases" using italics (line 315).
Response 80: Thank you for your suggestion. We formatted the "N" in "N-acetylglucosaminyltransferases" using italics.
Comment 81: Please change "progressive supranuclear palsy (PSP) and corticobasal degeneration (CBD)" to "PSP and CBD" (line 323).
Response 81: Thank you for your suggestion. We changed "progressive supranuclear palsy (PSP) and corticobasal degeneration (CBD)" to "PSP and CBD".
Comment 82: Please format "N" in "β-N-acetylglucosaminidase" using italics (line 330).
Response 82: Thank you for your suggestion. We formatted the "N" in "β-N-acetylglucosaminidase" using italics.
Comment 83: Please replace "facilitating" with "facilitating its" (line 331).
Response 83: Thank you for your suggestion. We replaced "facilitating" with "facilitating its".
Comment 84: Please change "plasticity; their" to "plasticity. Their" (line 343).
Response 84: Thank you for your suggestion. We changed "plasticity; their" to "plasticity. Their".
Comment 85: Please replace "RAGE" with "and RAGE" (lines 344, 468).
Response 85: Thank you for your suggestion. We replaced "RAGE" with "and RAGE".
Comment 86: Please change "hexosamine biosynthetic pathway (HBP)" to "HBP" (lines 346, 357).
Response 86: Thank you for your suggestion. We changed "hexosamine biosynthetic pathway (HBP)" to "HBP".
Comment 87: The two lines representing the arrow leading from "Tau N-glycosylation" seem to be disconnected at the corner in Figure 1. Please fix.
Response 87: Thank you for your suggestion. We fixed this issue in Figure 1.
Comment 88: Similarly, the two lines representing the arrow leading from "Tau O-GlcNAcylation" seem to be disonnected at the corner in Figure 1. Please fix.
Response 88: Thank you for your suggestion. We fixed this issue in Figure 1.
Comment 89: Please replace "APP Glycosylation" with "APP glycosylation", "RAGE Glycosylation" with "RAGE glycosylation", "Beta Amyloid Fibres" with "Amyloid beta fibrils", "Amyloid Beta Plaque" with "Amyloid beta plaque" in Figure 1.
Response 89: Thank you for your suggestion. We replaced "APP Glycosylation" with "APP glycosylation", "RAGE Glycosylation" with "RAGE glycosylation", "Beta Amyloid Fibres" with "Amyloid beta fibrils", "Amyloid Beta Plaque" with "Amyloid beta plaque" in Figure 1.
Comment 90: Please provide abbreviation for "HBP", "OGT", "OGA", "GnTs" in the legend to Figure 1.
Response 90: Thank you for your suggestion. We provided abbreviation for "HBP", "OGT", "OGA", "GnTs" in the legend to Figure 1.
Comment 91: From "Schematic Overview of Glycosylation in Alzheimer’s Disease and Tauopathies, illustrating how glycosylation affects tau, APP/Aβ, synaptic and immune receptors, and the enzymatic/metabolic regulators (HBP, OGT, OGA, GnTs)" (line 351) is not unequivocally clear whether the authors mean to say "enzymatic and metabolic regulators" or "enzymatic or metabolic regulators"? Please fix.
Response 91: Thank you for your suggestion. We changed "enzymatic/metabolic regulators" to "enzymatic and metabolic regulators".
Comment 92: Please change "Overview of Glycosylation in Alzheimer’s Disease and Tauopathies" to "overview of glycosylation in Alzheimer’s disease and tauopathies" (line 351).
Response 92: Thank you for your suggestion. We changed "Overview of Glycosylation in Alzheimer’s Disease and Tauopathies" to "overview of glycosylation in Alzheimer’s disease and tauopathies".
Comment 93: Please replace "synaptic" with "synaptic," (line 352).
Response 93: Thank you for your suggestion. We replaced "synaptic" with "synaptic,".
Comment 94: Please change "GnTs" to "and GnTs" (line 353).
Response 94: Thank you for your suggestion. We changed "GnTs" to "and GnTs".
Comment 95: Please remove bold formatting from "Created with bioRender." (line 353).
Response 95: Thank you for your suggestion. We removed bold formatting from "Created with bioRender.".
Comment 96: Please replace "Created with bioRender. Acronyms in figure defined: APP (amyloid precursor protein), RAGE (receptor for advanced glycation end products)" with "APP, amyloid precursor protein; RAGE, receptor for advanced glycation end products. Created with BioRender" (line 353).
Response 96: Thank you for your suggestion. We replace d"Created with bioRender. Acronyms in figure defined: APP (amyloid precursor protein), RAGE (receptor for advanced glycation end products)" with "APP, amyloid precursor protein; RAGE, receptor for advanced glycation end products. Created with BioRender".
Comment 97: Please format "N" in "UDP-N-acetylglucosamine" using italics (line 357).
Response 97: Thank you for your suggestion. We formatted the "N" in "UDP-N-acetylglucosamine" using italics.
Comment 98: Please change "glycogen synthase kinase-3β (GSK3β)" to "GSK3β" (line 369).
Response 98: Thank you for your suggestion. We changed "glycogen synthase kinase-3β (GSK3β)" to "GSK3β".
Comment 99: Please replace "Sialidase inhibitors, by preserving polysialylation, may" with "By preserving polysialylation, sialidase inhibitors may" (line 384).
Response 99: Thank you for your suggestion. We replaced "Sialidase inhibitors, by preserving polysialylation, may" with "By preserving polysialylation, sialidase inhibitors may".
Comment 100: Please replace "Alzheimer’s Disease" with "AD" (lines 387, 508).
Response 100: Thank you for your suggestion. We replaced "Alzheimer’s Disease" with "AD".
Comment 101: Please change "Glycosylation Processes in Alzheimer’s Disease and Tauopathies" to "glycosylation processes in Alzheimer’s disease and tauopathies" (line 389).
Response 101: Thank you for your suggestion. We changed "Glycosylation Processes in Alzheimer’s Disease and Tauopathies" to "glycosylation processes in Alzheimer’s disease and tauopathies".
Comment 102: Please replace "Acronyms in table defined: APP" with "APP" (line 390).
Response 102: Thank you for your suggestion. We replaced "Acronyms in table defined: APP" with "APP".
Comment 103: Please sort abbreviations according to the alphabetical order in Table 1.
Response 103: Thank you for your suggestion. We sorted abbreviations in Table 1 in alphabetical order.
Comment 104: Please change "APP (amyloid precursor protein)," to "APP, amyloid precursor protein;" (line 390).
Response 104: Thank you for your suggestion. We changed "APP (amyloid precursor protein)," to "APP, amyloid precursor protein;".
Comment 105: Please replace "Aβ (amyloid-β)," with "Aβ, amyloid-β;" (line 390).
Response 105: Thank you for your suggestion. We replaced "Aβ (amyloid-β)," with "Aβ, amyloid-β;".
Comment 106: Please change "NFT (neurofibrillary tangle)," to "NFT, neurofibrillary tangle;" (line 390).
Response 106: Thank you for your suggestion. We changed "NFT (neurofibrillary tangle)," to "NFT, neurofibrillary tangle;".
Comment 107: Please replace "NMDA (N-methyl-D-aspartate receptor)," with "NMDA, N-methyl-D-aspartate receptor;" (line 391).
Response 107: Thank you for your suggestion. We replaced "NMDA (N-methyl-D-aspartate receptor)," with "NMDA, N-methyl-D-aspartate receptor;".
Comment 108: Please change "AMPA (α-amino-3-hydroxy-5-methyl-4-isoxazolepropionic acid receptor)," to "AMPA, α-amino-3-hydroxy-5-methyl-4-isoxazolepropionic acid receptor;" (line 391).
Response 108: Thank you for your suggestion. We changed "AMPA (α-amino-3-hydroxy-5-methyl-4-isoxazolepropionic acid receptor)," to "AMPA, α-amino-3-hydroxy-5-methyl-4-isoxazolepropionic acid receptor;".
Comment 109: Please replace "NCAM (neural cell adhesion molecule)," with "NCAM, neural cell adhesion molecule;" (line 392).
Response 109: Thank you for your suggestion. We replaced "NCAM (neural cell adhesion molecule)," with "NCAM, neural cell adhesion molecule;".
Comment 110: Please change "TREM2 (triggering receptor expressed on myeloid cells 2)," to "TREM2, triggering receptor expressed on myeloid cells 2;" (line 392).
Response 110: Thank you for your suggestion. We changed "TREM2 (triggering receptor expressed on myeloid cells 2)," to "TREM2, triggering receptor expressed on myeloid cells 2;".
Comment 111: Please replace "CD33 (cluster of differentiation 33)," with "CD33, cluster of differentiation 33;" (line 393).
Response 111: Thank you for your suggestion. We replaced "CD33 (cluster of differentiation 33)," with "CD33, cluster of differentiation 33;".
Comment 112: Please change "IL-6 (interleukin-6)," to "IL-6, interleukin-6;" (line 393).
Response 112: Thank you for your suggestion. We change "IL-6 (interleukin-6)," to "IL-6, interleukin-6;".
Comment 113: Please replace "TNF-α (tumor necrosis factor-alpha)," with "TNF-α, tumor necrosis factor-alpha;" (line 393).
Response 113: Thank you for your suggestion. We replaced "TNF-α (tumor necrosis factor-alpha)," with "TNF-α, tumor necrosis factor-alpha;".
Comment 114: Please change "CSF (cerebrospinal fluid)," to "CSF, cerebrospinal fluid;" (line 394).
Response 114: Thank you for your suggestion. We changed "CSF (cerebrospinal fluid)," to "CSF, cerebrospinal fluid;".
Comment 115: Please replace "OGT (O-GlcNAc transferase)," with "OGT, O-GlcNAc transferase;" (line 394).
Response 115: Thank you for your suggestion. We replace "OGT (O-GlcNAc transferase)," with "OGT, O-GlcNAc transferase;".
Comment 116: Please change "OGA (O-GlcNAcase)" to "OGA, O-GlcNAcase" (line 394).
Response 116: Thank you for your suggestion. We changed "OGA (O-GlcNAcase)" to "OGA, O-GlcNAcase".
Comment 117: Please replace "“common denominator.”" with "“common denominator”." (line 408).
Response 117: Thank you for your suggestion. We replaced "“common denominator.”" with "“common denominator”.".
Comment 118: Please change "it" to "which" (line 411).
Response 118: Thank you for your suggestion. We changed "it" to "which".
Comment 119: Please replace "strategies, interventions" with something like "strategies. Hence, interventions" (line 415).
Response 119: Thank you for your suggestion. We replace "strategies, interventions" with something like "strategies. Hence, interventions".
Comment 120: Please change "hexosamine biosynthetic pathway" to "HBP" (lines 420, 471).
Response 120: Thank you for your suggestion. We changed "hexosamine biosynthetic pathway" to "HBP".
Comment 121: Please replace "stability; mutations" with "stability. Mutations" (line 428).
Response 121: Thank you for your suggestion. We replaced "stability; mutations" with "stability. Mutations".
Comment 122: Please change "remain: glycan" to "remain. Glycan" (line 439).
Response 122: Thank you for your suggestion. We changed "remain: glycan" to "remain. Glycan".
Comment 123: Please replace "several" with "the following" (line 458).
Response 123: Thank you for your suggestion. We replaced "several" with "the following".
Comment 124: Please change "emerge:" to "emerge" (line 459).
Response 124: Thank you for your suggestion. We changed "emerge:" to "emerge".
Comment 125: Please replace "Tau regulation: Aberrant" with "Aberrant" (line 460).
Response 125: Thank you for your suggestion. We replaced "Tau regulation: Aberrant" with "Aberrant".
Comment 126: Please change "aggregation, while" to "aggregation while" (line 461).
Response 126: Thank you for your suggestion. We changed "aggregation, while" to "aggregation while".
Comment 127: Please replace "Amyloid precursor protein (APP) and amyloid-β (Aβ): N-" with "N-" (line 463).
Response 127: Thank you for your suggestion. We replaced "Amyloid precursor protein (APP) and amyloid-β (Aβ): N-" with "N-".
Comment 128: Please format "N" in "N-" using italics (line 463).
Response 128: Thank you for your suggestion. We formatted the "N" in "N-" using italics.
Comment 129: Please change "Synaptic and immune regulation: Glycosylation" to "Glycosylation" (line 466).
Response 129: Thank you for your suggestion. We changed "Synaptic and immune regulation: Glycosylation" to "Glycosylation".
Comment 130: Please replace "Enzymatic control and metabolism: Enzymes" with "Enzymes" (line 469).
Response 130: Thank you for your suggestion. We replaced "Enzymatic control and metabolism: Enzymes" with "Enzymes".
Comment 131: Please change "Biomarker potential: Glycoproteomic" to "Glycoproteomic" (line 473).
Response 131: Thank you for your suggestion. We changed "Biomarker potential: Glycoproteomic" to "Glycoproteomic".
Comment 132: Please replace "Directions" with "directions" (line 479).
Response 132: Thank you for your suggestion. We replaced "Directions" with "directions".
Comment 133: Please change "remain:" to "remain." (line 481).
Response 133: Thank you for your suggestion. We changed "remain:" to "remain."
Comment 134: Please replace "Clarifying temporal dynamics: Longitudinal" with "Longitudinal" (line 482).
Response 134: Thank you for your suggestion. We replaced "Clarifying temporal dynamics: Longitudinal" with "Longitudinal".
Comment 135: Please change "Expanding glycoproteomic profiling: High-resolution" to "High-resolution" (line 486).
Response 135: Thank you for your suggestion. We changed "Expanding glycoproteomic profiling: High-resolution" to "High-resolution".
Comment 136: Please replace "Targeted therapeutics: Clinical" with "Clinical" (line 490).
Response 136: Thank you for your suggestion. We replace "Targeted therapeutics: Clinical" with "Clinical".
Comment 137: Please change "cautiously, with" to "cautiously with" (line 491).
Response 137: Thank you for your suggestion. We changed "cautiously, with" to "cautiously with".
Comment 138: Please replace "glycosylation (protective vs. pathogenic)" with "glycosylation, protective vs. pathogenic" (line 492).
Response 138: Thank you for your suggestion. We replaced "glycosylation (protective vs. pathogenic)" with "glycosylation, protective vs. pathogenic".
Comment 139: Please change "Systems biology approaches: Glycosylation" to "Glycosylation" (line 495).
Response 139: Thank you for your suggestion. We changed "Systems biology approaches: Glycosylation" to "Glycosylation".
Comment 140: Please replace "Expanding to non-AD tauopathies: Comparative" with "Comparative" (line 499).
Response 140: Thank you for your suggestion. We replace "Expanding to non-AD tauopathies: Comparative" with "Comparative".
Comment 141: Please change "Bridging bench to bedside: Translation" to "Translation" (line 503).
Response 141: Thank you for your suggestion. We changed "Bridging bench to bedside: Translation" to "Translation".
Comment 142: Please replace "processing, while" with "processing while" (line 521).
Response 142: Thank you for your suggestion. We replaced "processing, while" with "processing while" (line 521).
Comment 143: Please change "cerebrospinal fluid" to "CSF" (line 527).
Response 143: Thank you for your suggestion. We changed "cerebrospinal fluid" to "CSF".
Comment 144: Please replace "M.C." with "M.C.," (lines 536, 537).
Response 144: Thank you for your suggestion. We replaced "M.C." with "M.C.,".
Comment 145: Please change "M.B., T.C." to "M.B., and T.C." (lines 537, 538, 539 2x, 540, 541, 542).
Response 145: Thank you for your suggestion. We changed "M.B., T.C." to "M.B., and T.C.".
Comment 146: Please replace "M.B. and" with "M.B., and" (line 537).
Response 146: Thank you for your suggestion. We replaced "M.B. and" with "M.B., and".
Comment 147: Please change "investigation, A.C.B., M.P.I., M.C., A.B." to "investigation, A.C.B., M.P.I., M.C., and A.B." (line 538).
Response 147: Thank you for your suggestion. We changed "investigation, A.C.B., M.P.I., M.C., A.B." to "investigation, A.C.B., M.P.I., M.C., and A.B.".
Comment 148: Please replace "A.C.B., M.P.I." with "A.C.B. and M.P.I." (line 539).
Response 148: Thank you for your suggestion. We replaced "A.C.B., M.P.I." with "A.C.B. and M.P.I.".
Comment 149: Please change "editing, M.C., A.B." to "editing, M.C. and A.B." (line 540).
Response 149: Thank you for your suggestion. We changed "editing, M.C., A.B." to "editing, M.C. and A.B.".
Comment 150: Please replace "M.P.I., M.C., A.B." with "M.P.I., M.C., and A.B." (line 540).
Response 150: Thank you for your suggestion. We replaced "M.P.I., M.C., A.B." with "M.P.I., M.C., and A.B.".
We thank you for all your comments and suggestions. We have addressed all 153 of them accordingly and we consider they helped a great deal in substantially increasing the overall value of our manuscript.

Reviewer 4 Report
Comments and Suggestions for Authors
This article presents a thorough and up-to-date analysis of the growing evidence implicating glycosylation in the pathogenesis of Alzheimer’s disease (AD) and associated tauopathies. The authors successfully integrate diverse research findings, bridging molecular mechanisms, neuropathological features, and clinical relevance. A major strength lies in the detailed exploration of N-linked and O-linked glycosylation as central modulators of protein misfolding, degradation, and neuroinflammatory responses. Moreover, the review highlights the potential utility of glycosylation-related markers in diagnosis and therapy, thereby increasing its translational importance. By incorporating cutting-edge developments in glycoproteomic technologies and examining the dynamic interplay between glycosylation and phosphorylation, the work adopts a progressive and integrative approach. Collectively, this study fills an essential void in existing literature by organizing a highly intricate and fast-moving research domain into a clear and thoughtfully constructed framework.
Suggestions for improvement:
(1) Despite its scientific depth, the use of technical terminology—including "tauopathies," "glycosyltransferases," and "glycoproteomics"—may hinder understanding for readers outside the field. Introducing brief definitions or contextual explanations early in the manuscript could improve readability without compromising scholarly quality.
(2) The relationship between aberrant glycosylation and observable clinical outcomes—such as progression of cognitive impairment or staging of disease severity—needs deeper discussion. Including empirical data from long-term cohort investigations or biomarker assessments in CSF would better connect biochemical changes to clinical reality.
(3) While AD, FTLD, PSP, and CBD are acknowledged, the distinctions in glycosylation profiles across these conditions remain underdeveloped. A comparative summary table outlining glycosylation changes specific to each disorder would improve clarity and emphasize unique pathological signatures.
(4) Technical hurdles in glycoproteomics, particularly in mapping precise glycosylation sites, continue to pose challenges. A dedicated discussion on current analytical limitations and emerging technological solutions would contribute to a more nuanced and objective appraisal.
(5) Although the written content is comprehensive, visual tools could greatly aid understanding. Incorporating a diagram illustrating the interactions between glycosylation, phosphorylation, tau pathology, and inflammation, as well as a graphical overview of therapeutic interventions targeting glycosylation, would strengthen conceptual communication and practical applicability.
Author Response
Comment 1: Despite its scientific depth, the use of technical terminology—including "tauopathies," "glycosyltransferases," and "glycoproteomics"—may hinder understanding for readers outside the field. Introducing brief definitions or contextual explanations early in the manuscript could improve readability without compromising scholarly quality.
Response 1: Thank you for your suggestions. We have create new explanatory paragraphs that we have inserted in the manuscript's text, in the Introductions, Section 2 and Section 5. We consider this increased the availability of our paper for general readers.
Comment 2: The relationship between aberrant glycosylation and observable clinical outcomes—such as progression of cognitive impairment or staging of disease severity—needs deeper discussion. Including empirical data from long-term cohort investigations or biomarker assessments in CSF would better connect biochemical changes to clinical reality.
Response 2: Thank you for your suggestion. We have made the necessary adjustments by creating subsection 6.6. Clinical correlations of Tau glycosylation with cognitive decline and disease staging.
Comment 3: While AD, FTLD, PSP, and CBD are acknowledged, the distinctions in glycosylation profiles across these conditions remain underdeveloped. A comparative summary table outlining glycosylation changes specific to each disorder would improve clarity and emphasize unique pathological signatures.
Response 3: Thank your for your suggestion. We have created subsection 2.5. Comparative Analysis of Tau Glycosylation and Phosphorylation Sites accordingly. Also we have created the table at the end of subsection 6.6.
Comment 4: Technical hurdles in glycoproteomics, particularly in mapping precise glycosylation sites, continue to pose challenges. A dedicated discussion on current analytical limitations and emerging technological solutions would contribute to a more nuanced and objective appraisal.
Response 4: Thank your for your suggestion. We have created subsection 5.7. to have a dedicated disscusion on current limitations and emerging solutions.
Comment 5: Although the written content is comprehensive, visual tools could greatly aid understanding. Incorporating a diagram illustrating the interactions between glycosylation, phosphorylation, tau pathology, and inflammation, as well as a graphical overview of therapeutic interventions targeting glycosylation, would strengthen conceptual communication and practical applicability.
Response 5: Thank your for your comment. We have created a figure illustrating the interactions between glycosylation, phosphorylation, tau pathology and inflammation that we have inserted in subsection 7.3. We consider this helped us increase the overall quality of our manuscript.

Round 2
Reviewer 2 Report
Comments and Suggestions for Authors
The authors have provided sufficient evidence to address my original comments, I am now satisfied with it and have no further comments.
Author Response
Comment: The authors have provided sufficient evidence to address my original comments, I am now satisfied with it and have no further comments.
Response: Thank you for your valuable input, which has helped us strengthen the scientific quality of our manuscript.
Reviewer 3 Report
Comments and Suggestions for Authors
Major points:
1) The "5.7. Analytical Limitations and Emerging Solutions in Glycoproteomics" chapter seems to bear rather a tabulated structure, despite being formulated as a general text. Instead, the authors may wish to consider converting the whole chapter into a new table with references (Table 1).
2) The same holds true for the "7.8. Glycosylation-Focused Guidelines for Tauopathies: Toward Consensus on Tau Pathology" chapter. Please organize the content of this chapter graphically as a new table.
Minor points:
1) Please provide reference for "Clinically, AD is defined by progressive cognitive decline, memory impairment, and eventual loss of autonomy" (line 43).
2) Please provide reference for "A crucial layer of regulation in these processes is post-translational modification (PTM), which fine-tunes protein structure, stability, and localization" (line 49).
3) Please provide reference for "Glycosylation influences protein folding, trafficking, degradation, and cell–cell signal-58 ling, processes that are fundamental to neuronal viability" (line 58).
4) Please provide reference for "Tauopathies, a group of neurodegenerative diseases characterized by abnormal aggregation of the microtubule-associated protein Tau, include AD, progressive supranuclear palsy (PSP), corticobasal degeneration (CBD), and Pick’s disease" (line 60).
5) Please change "Tau, include" to "tau include" (line 61).
6) Please provide reference for "These disorders share Tau accumulation but differ in isoform composition, filament morphology, and regional pathology" (line 62).
7) Please replace "Tau" with "tau" (lines 63, 99, 148, 165, 167, 173, 176, 312, 344, 350, 352, 355, 364, 397, 425, 506, 507, 509, 512, 515, 522, 523, 573, 629, 640, 673).
8) Please provide reference for "In tauopathies such as frontotemporal lobar degeneration (FTLD), progressive supranuclear palsy (PSP), and corticobasal degeneration (CBD), glycosylation abnormalities intersect with tau phosphorylation, altering aggregation dynamics and neuronal toxicity" (line 65).
9) Please change "progressive supranuclear palsy (PSP)" to "PSP" (lines 65).
10) Please replace "corticobasal degeneration (CBD)" with "CGD" (line 66).
11) Please change "[3]" to "[3]." (line 69).
12) Please provide reference for "Moreover, glycosylation affects receptor signalling, immune activation, and synaptic function, positioning it as a central regulator of neurodegeneration." (line 69).
13) Please replace "hexosamine biosynthetic pathway" with "hexosamine biosynthetic pathway (HBP)" (lines 76, 110, 435) and "hexosamine biosynthetic pathway (HBP)" with "HBP" (line 435).
14) Please replace "Alzheimer’s" with "Alzheimer’s disease" (line 82).
15) Please provide reference for "Tau protein is a microtubule-associated protein essential for stabilizing cytoskeletal architecture and supporting axonal transport" (line 83).
16) Please change "tau’s conformation" to "the conformation of tau" (line 91).
17) Please provide reference for "Importantly, aberrant N-glycosylation often precedes or facilitates hyperphosphorylation" (line 94).
18) Please merge "Glycosyltransferases—enzymes that catalyze the transfer of sugar moieties to proteins or lipids—mediate the covalent addition of glycan structures that can modulate Tau folding, stability, and aggregation" (line 98) with the previous paragraph.
19) Please provide reference for "Glycosyltransferases—enzymes that catalyze the transfer of sugar moieties to proteins or lipids—mediate the covalent addition of glycan structures that can modulate Tau folding, stability, and aggregation" (line 98).
20) Please provide reference for "While all tauopathies share abnormal tau aggregation, key differences exist in isoform composition, filament structure, and regional vulnerability" (line 120).
21) Please provide reference for "In AD, neurofibrillary tangles contain both 3-repeat (3R) and 4-repeat (4R) tau isoforms that assemble into paired helical filaments (PHFs), typically distributed from the entorhinal cortex to the neocortex in a hierarchical pattern" (line 121).
22) Please provide reference for "In contrast, PSP and CBD predominantly accumulate 4R-tau, forming straight filaments and involving basal ganglia, brainstem, and motor cortex" (line 124).
23) Please provide reference for "Frontotemporal lobar degeneration (FTLD-tau) subtypes vary—Pick’s disease features 3R-tau inclusions (Pick bodies) within frontotemporal regions, whereas globular glial tauopathies exhibit mixed isoforms with astrocytic or oligodendroglial predominance" (line 126).
24) Please replace "Frontotemporal lobar degeneration (FTLD-tau)" with "FTLD-tau" (line 126).
25) Please provide reference for "Evidence indicates that AD-associated tau tends to be more extensively N-glycosylated and exhibits reduced O-GlcNAcylation, while PSP and CBD display more restricted N-glycan complexity and relatively conserved O-GlcNAc profiles" (line 130).
26) Please change "Analysis of Tau Glycosylation and Phosphorylation Sites" to "analysis of tau glycosylation and phosphorylation sites" (line 147).
27) Please provide reference for "N-glycosylation appears to promote Tau hyperphosphorylation and aggregation, whereas O-GlcNAcylation exerts a protective role" (line 148).
28) Please provide reference for "Tau protein contains over 80 potential phosphorylation sites and at least 10 experimentally confirmed glycosylation sites, many of which are spatially proximal or overlapping" (line 151).
29) Please provide reference for "N-glycosylation typically occurs on asparagine residues (e.g., Asn167, Asn359), whereas O-GlcNAcylation affects serine and threonine residues (e.g., Ser400, Thr123, Thr245, Ser356) that often coincide with major phosphorylation motifs recognized by kinases such as GSK3β, CDK5, and MARK2" (line 153).
30) Please define abbreviation for "CDK5" (line 156), "MARK2" (line 156), "HILIC" (line 403), "FDR" (line 409), "ETD" in "ETD/EThcD/AI-ETD" (line 417), "EThcD" in "ETD/EThcD/AI-ETD" (line 417), "AI-ETD" in "ETD/EThcD/AI-ETD" (line 417), "HCD" in "stepped-HCD" (line 417), "FAIMS" in "FAIMS/TIMS-PASEF" (line 419), "TIMS-PASEF" in "FAIMS/TIMS-PASEF" (line 419), "PRM" in "PRM/DIA-SWATH" (line 422), "DIA-SWATH" in "PRM/DIA-SWATH" (line 422), "IMS" (line 425), "LoD" in "LoD/LoQ" (line 650), "LoQ" in "LoD/LoQ" (line 650), "AT8" in "AT8/PHF-1/CP13" (line 661), "PHF-1" in "AT8/PHF-1/CP13" (line 661), "CP13" in "AT8/PHF-1/CP13" (line 661), "PNGase" (line 664), "IP" in "IP→LC-MS/MS" (line 667).
31) Please provide reference for "These modifications act in a competitive and regulatory manner. N-glycosylation changes Tau’s tertiary structure, reducing its microtubule affinity and exposing neighboring serine/threonine residues to kinases, thereby facilitating hyperphosphorylation" (line 157).
32) From "These modifications act in a competitive and regulatory manner. N-glycosylation changes Tau’s tertiary structure, reducing its microtubule affinity and exposing neighboring serine/threonine residues to kinases, thereby facilitating hyperphosphorylation" (line 157) is not unequivocally clear whether the authors mean to say "serine and threonine residues" or "serine or threonine residues"? Please fix.
33) Please merge "These modifications act in a competitive and regulatory manner. N-glycosylation changes Tau’s tertiary structure, reducing its microtubule affinity and exposing neighboring serine/threonine residues to kinases, thereby facilitating hyperphosphorylation. In contrast, O-GlcNAcylation directly competes with phosphorylation at the same or adjacent residues, inhibiting excessive phosphorylation and stabilizing Tau–microtubule interactions. Consequently, decreased O-GlcNAcylation in AD — often linked to impaired glucose metabolism — removes this protective brake, leading to pathological phosphorylation and tangle formation" (line 157) with the previous paragraph.
34) Please replace "Tau’s tertiary structure" with "the tertiary structure of tau" (line 158).
35) Please provide reference for "In contrast, O-GlcNAcylation directly competes with phosphorylation at the same or adjacent residues, inhibiting excessive phosphorylation and stabilizing Tau–microtubule interactions" (line 160).
36) Please change "Tau–microtubule" to "T
tau–microtubule" (line 161).
37) Please provide reference for "Consequently, decreased O-GlcNAcylation in AD — often linked to impaired glucose metabolism — removes this protective brake, leading to pathological phosphorylation and tangle formation" (line 162).
38) Please remove the gap between lines 164 and 165.
39) Please provide reference for "In AD, both 3R and 4R Tau isoforms exhibit extensive N-glycosylation and reduced O-GlcNAcylation, driving PHF assembly" (line 165).
40) Please provide reference for "In other tauopathies such as PSP and CBD, Tau aggregates are composed predominantly of 4R isoforms with more restricted glycosylation patterns and distinct phosphorylation site usage (e.g., Ser262, Ser396)" (line 166).
41) Please remove the gap between lines 171 and 172.
42) Please provide reference for "Future work should address these site-specific interactions, emphasizing that N-glycosylation (Asn167/359) acts as an early structural modifier that primes Tau for pathogenic phosphorylation, while O-GlcNAcylation (Ser400, Thr245, Ser356) counteracts aggregation by masking phosphorylation motifs" (line 172).
43) Please replace "amyloid precursor protein (APP)" with "APP" (line 179).
44) Please change "in" to "in the" (line 187).
45) Please provide reference for "In addition to N-glycosylation, APP and its secretases undergo O-glycosylation, particularly O-GalNAc modification, which influences protein trafficking and enzymatic activity" (line 195).
46) Please provide reference for "BACE1 contains multiple N-glycans that regulate folding, transport, and stability" (line 200).
47) Please provide reference for "Similarly, γ-secretase components such as nicastrin are highly glycosylated" (line 202).
48) Please provide reference for "These modifications fine-tune Aβ species ratios, including the pathogenic Aβ42/40 balance" (line 204).
49) Please provide reference for "Modified Aβ shows altered aggregation kinetics and toxicity" (line 208).
50) Please provide reference for "Aberrant glycosylation of synaptic proteins, receptors, and ion channels exacerbates Aβ-induced dysfunction" (line 218).
51) Please format "N" in "N-methyl-d-aspartate" using italics (line 219).
52) Please format "d" in "N-methyl-d-aspartate" using small caps (line 219).
53) Please provide reference for "Furthermore, microglial and astrocytic receptors involved in Aβ clearance—such as triggering receptor expressed on myeloid cells 2 (TREM2) and Cluster of Differentiation 33 (CD33)—are heavily glycosylated" (line 224).
54) Please replace "Cluster of Differentiation" with "cluster of differentiation" (line 225).
55) Please change "vasculature." to "vasculature" (line 238).
56) Please provide reference for "In AD models, aberrant glycosylation disrupts NMDA receptor localization, sensitizing neurons to Aβ-induced excitotoxicity" (line 247).
57) Please provide reference for "Similarly, α-amino-3-hydroxy-5-methyl-4-isoxazolepropionic acid (AMPA) receptors rely on N-glycans for stability and gating properties" (line 250).
58) Please remove bold formatting from "(" in "(AMPA)" (line 250).
59) Please provide reference for "Adhesion molecules such as neural cell adhesion molecule (NCAM) and L1CAM are highly glycosylated, and polysialylation of NCAM regulates neurite outgrowth and synaptic remodeling" (line 254).
60) Please provide reference for "TREM2 contains multiple N-glycosylation sites that regulate folding and surface expression" (line 260).
61) Please replace "The CD33" with "CD33" (line 265).
62) Please remove bold formatting from "(" in "(GFAP)-interacting" (line 269).
63) Please provide reference for "Interleukin-6 (IL-6), a pro-inflammatory cytokine elevated in AD, requires N-glycosylation for secretion and receptor binding" (line 273).
64) Please provide reference for "Chemokine receptors such as CCR5 and CX3CR1 are also glycosylated, with changes in glycan branching modulating microglial migration and response to injury" (line 278).
65) Please change "chronic, non-resolving" to "chronic non-resolving" (line 280).
66) Please provide reference for "Glycosylation also extends its influence to pattern recognition receptors (PRRs) that detect damage-associated molecular patterns (DAMPs)" (line 292).
67) Please provide reference for "In addition, toll-like receptors (TLRs) rely on N-glycans for stability and ligand recognition" (line 297).
68) Please replace "Alzheimer’s disease (AD)" with "AD" (lines 306, 545).
69) Please provide reference for "Glycoproteomics provides detailed mapping of glycan sites and structures, enabling the identification of disease-specific Tau modifications" (line 311).
70) Please change "bridging" to something like "thereby bridging" (line 340).
71) Please provide reference for "These tools have revealed regional differences in glycosylation within hippocampus, cortex, and subcortical structures, aligning with clinical phenotypes of memory loss, executive dysfunction, and motor impairment" (line 340).
72) Please provide reference for "Most commercially available anti-Tau antibodies are designed against phosphorylated epitopes (e.g., pSer202, pThr231, pSer396), many of which have been extensively validated for immunohistochemistry (IHC), Western blot, and enzyme-linked immunosorbent assay (ELISA) applications in brain and CSF samples" (line 345).
73) Please replace "anti-Tau" with "anti-tau" (line 346).
74) Please provide reference for "In contrast, antibodies specifically targeting glycosylated Tau are scarce and often lack full validation due to the structural diversity and low abundance of glycoforms" (line 349).
75) Please provide reference for "Validation of Tau glycosylation-specific antibodies for clinical biofluids such as CSF or serum remains preliminary, primarily constrained by low antigen concentration and conformational masking of glycan epitopes" (line 355).
76) Please provide reference for "While recent studies have employed antibody–lectin hybrid assays and antibody enrichment followed by LC–MS/MS to enhance specificity, no single antibody currently demonstrates broad cross-platform reliability" (line 357).
77) Please provide reference for "For immunohistochemical detection, phospho-Tau antibodies remain highly reliable and routinely used to stage pathology (e.g., AT8, PHF-1, CP13), whereas glycosylation-targeting antibodies are still in experimental phases" (line 359).
78) Please change "immunohistochemical" to "IHC" (line 360).
79) Please replace "phospho-Tau" with "phospho-tau" (lines 360).
80) Please change "identifies" to "identified" (line 375).
81) Please provide reference for "Glycan heterogeneity 386 complicates quantification, and standardized reference libraries are limited" (line 386).
82) Please replace "Longitudinal" with "In addition, longitudinal" (line 392).
83) Please justify the text presented in the "5.7. Analytical Limitations and Emerging Solutions in Glycoproteomics" chapter.
84) Please change "Limitations and Emerging Solutions in Glycoproteomics" to "limitations and emerging solutions in glycoproteomics" (line 395).
85) Please replace "Current limitations. Despite" with "Despite" (line 396).
86) Please provide reference for "(i) Microheterogeneity and isomerism: a single glycosite carries multiple glycoforms, and many glycans are isomeric, making confident assignment difficult" (line 398).
87) Please change "(i) Microheterogeneity and isomerism: a" to "A" (line 398).
88) Please provide reference for "(ii) Fragmentation bias and site localization: collision-based fragmentation can preferentially cleave labile glycosidic bonds, yielding rich glycan information but poor peptide backbone coverage; this limits exact site localization on multiply modified peptides" (line 399).
89) Please replace "(ii) Fragmentation bias and site localization: collision-based" with "Collision-based" (line 399).
90) Please change "coverage; this" to "coverage. This" (line 402).
91) Please provide reference for "(iii) Enrichment bias and coverage: lectin/HILIC enrichments favor subsets of glycans, under-representing low-abundance or highly sialylated/fucosylated species; co-purified non-glycopeptides reduce dynamic range" (line 402).
92) From "(iii) Enrichment bias and coverage: lectin/HILIC enrichments favor subsets of glycans, under-representing low-abundance or highly sialylated/fucosylated species; co-purified non-glycopeptides reduce dynamic range" (line 402) is not unequivocally clear whether the authors mean to say "highly sialylated and fucosylated species" or "highly sialylated or fucosylated species"? Please correct.
93) Please replace "(iii) Enrichment bias and coverage: lectin/HILIC" with "Lectin/HILIC" (line 402).
94) Please change "species; co-purified" to "species. Co-purified" (line 404).
95) Please provide reference for "(iv) Quantification and comparability: most datasets are semi-quantitative; absolute quantification is hindered by the lack of robust isotopically labeled glycopeptide standards, matrix effects in CSF/plasma, batch effects, and inter-laboratory variability" (line 405).
96) From "(iv) Quantification and comparability: most datasets are semi-quantitative; absolute quantification is hindered by the lack of robust isotopically labeled glycopeptide standards, matrix effects in CSF/plasma, batch effects, and inter-laboratory variability" (line 405) is not unequivocally clear whether the authors mean to say "CSF and plasma" or "CSF or plasma"? Pleas fix.
97) Please replace "(iv) Quantification and comparability: most" with "Most" (line 405).
98) Please change "semi-quantitative; absolute" to "semi-quantitative. Absolute" (line 406).
99) Please provide reference for "(v) False discovery and informatics: assigning spectra to specific glyco-proteoforms increases the search space and FDR; peptide-centric pipelines often struggle with co-existing PTMs (e.g., O-GlcNAc/phosphorylation) and with reporting site occupancy" (line 408).
100) From "(v) False discovery and informatics: assigning spectra to specific glyco-proteoforms increases the search space and FDR; peptide-centric pipelines often struggle with co-existing PTMs (e.g., O-GlcNAc/phosphorylation) and with reporting site occupancy" (line 408) is not unequivocally clear whether the authors mean ty say " O-GlcNAc and phosphorylation" or " O-GlcNAc or phosphorylation"? Please correct.
101) Please replace "(v) False discovery and informatics: assigning" with "Assigning" (line 408).
102) Please change "FDR; peptide-centric" to "FDR. Peptide-centric" (line 409).
103) Please provide reference for "(vi) Sample handling artifacts: partial desialylation or de-N-glycosylation during preparation can distort native profiles; limited material from biofluids constrains replication" (line 411).
104) Please replace "(vi) Sample handling artifacts: partial" with "Partial" (line 411).
105) Please change "profiles; limited" to "profiles. Limited" (line 412).
106) Please provide reference for "(vii) Spatial context: bulk proteomics lacks regional and cell-type resolution needed to relate glycosylation to tauopathy topography" (line 413).
107) Please replace "(vii) Spatial context: bulk" with "Bulk" (line 413).
108) Please change "Emerging solutions. Multiple" to "Multiple" (line 416).
109) Please provide reference for "(a) Hybrid fragmentation (e.g., ETD/EThcD/AI-ETD or stepped-HCD) improves simultaneous backbone and glycan characterization, enhancing site localization" (line 416).
110) Please replace "(a) Hybrid" with "Hybrid" (line 416).
111) Please provide reference for "(b) Ion mobility (e.g., FAIMS/TIMS-PASEF) separates isomeric/isobaric glycopeptides, boosts sensitivity, and reduces chemical noise in CSF" (line 418).
112) From "(b) Ion mobility (e.g., FAIMS/TIMS-PASEF) separates isomeric/isobaric glycopeptides, boosts sensitivity, and reduces chemical noise in CSF" (line 418) is not unequivocally clear whether the authors mean to say "isomeric and isobaric glycopeptides" or "isomeric or isobaric glycopeptides" (line 418)? Please correct.
113) Please change "(b) Ion" to "Ion" (line 418).
114) Please provide reference for "(c) Orthogonal decoding using exoglycosidase arrays and derivatization (e.g., permethylation, linkage-specific sialic acid chemistry) constrains structural assignments" (line 420).
115) Please replace "(c) Orthogonal" with "(c) Orthogonal" (line 420).
116) Please provide reference for "(d) Targeted quantification with PRM/DIA-SWATH and stable-isotope internal standards enables reproducible, multi-center-ready panels" (line 422).
117) Please change "(d) Targeted" to "Targeted" (line 422).
118) Please provide reference for "(e) Top-down and native MS resolve intact proteoforms (co-occurring glyco- and phospho-states), clarifying crosstalk on Tau" (line 423).
119) Please change "reproducible, multi-center-ready" to "reproducible multi-center-ready" (line 423).
120) Please replace "(e) Top-down" with "Top-down" (line 423).
121) Please provide reference for "(f) Spatial glycoproteomics (MALDI imaging, lectin-guided IMS, laser capture microdissection → LC–MS) adds anatomical resolution aligned with Braak staging" (line 425).
122) From "Spatial glycoproteomics (MALDI imaging, lectin-guided IMS, laser capture microdissection → LC–MS) adds anatomical resolution aligned with Braak staging" (line 425) is not clear what the authors refer to as the "Braak staging"? Please corroborate in the text.
123) From "(g) Improved bioinformatics (open/offset searches, glycan-aware FDR control, site-occupancy reporting) and reporting standards (minimum metadata, performance metrics, enzyme-toggle controls) enhance transparency and comparability" (line 427) is not unequivocally clear whether the authors mean to say "open and offset searches" or "open or offset searches"? Please fix.
124) Please change "(f) Spatial" to "Spatial" (line 425).
125) Please replace "microdissection →" with "microdissection," (line 426).
126) Please change "(g) Improved" to "Improved" (line 427).
127) Please replace "Alzheimer’s disease" with "AD" (lines 436, 509).
128) Please replace "AD" with "Alzheimer’s disease" (line 440).
129) Please provide reference for "Abnormal N-glycosylation of tau promotes hyperphosphorylation and aggregation, while reduced O-GlcNAcylation removes a protective brake on phosphorylation" (line 463).
130) Please provide reference for "N- and O-glycosylation of APP and secretases regulate Aβ generation, with aberrant patterns favoring amyloidogenic processing" (line 465).
131) Please provide reference for "Synaptic receptors and adhesion molecules require proper glycosylation for stability and plasticity" (line 466).
132) Please provide reference for "Immune receptors (TREM2, CD33, and RAGE) and cytokines are glycosylated, shaping microglial and astrocytic activation, while complement protein glycosylation regulates synapse pruning" (line 468).
133) Please provide reference for "Finally, altered activity of glycosyltransferases, glycosidases, and the HBP underlies these shifts" (line 470).
134) "A diagram of a cell cycle AI-generated content may be incorrect." sign appears when hovering the mouse cursor over Figure 1. Please disable this feature.
135) From "Schematic overview of glycosylation in Alzheimer’s disease and tauopathies, illustrating how glycosylation affects tau APP/Aβ, synaptic, and immune receptors, and the enzymatic and metabolic regulators (HBP, OGT, OGA, and GnTs)" (line 475) is not unequivocally clear whether the authors mean to say "APP and Aβ" or "APP or Aβ"? Please correct.
136) Please change "bioRender" to "BioRender" (lines 477, 574, 636).
137) Please merge "APP, amyloid precursor protein; RAGE, receptor for advanced glycation end products; HBP, hexosamine biosynthetic pathway; OGT, O-GlcNAc transferase; OGA, O-GlcNAcase; GnTs, N-acetylglucosaminyltransferases" (line 478) with the legend to Figure 1.
138) Please sort "APP, amyloid precursor protein; RAGE, receptor for advanced glycation end products; HBP, hexosamine biosynthetic pathway; OGT, O-GlcNAc transferase; OGA, O-GlcNAcase; GnTs, N-acetylglucosaminyltransferases" (line 478) according to the alphabetical order.
139) Please provide reference for "Similarly, caloric restriction and metabolic interventions that enhance glucose utilization may indirectly restore protective glycosylation patterns" (line 489).
140) Please provide reference for "Aberrant expression of glycosylation enzymes is detectable in CSF and serum, offering biomarker potential" (line 501).
141) Please remove the gap between lines 511 and 512.
142) Please provide reference for "Conversely, reduced O-GlcNAcylation levels in Tau correlate with accelerated cognitive decline and lower Mini-Mental State Examination (MMSE) scores, suggesting a loss of neuroprotective regulation" (line 512).
143) Please remove the gap between lines 517 and 518.
144) Please replace "mass spectrometry" with "MS" (line 519).
145) Please change "tauopathies" to "tauopathies." (line 525).
146) Please sort "CSF, cerebrospinal fluid; NFT, neurofibrillary tangle; O-GlcNAc, O-linked N-acetylglucosamine; PHF, paired helical filament; 3R/4R, three/four microtubule-binding repeats" (line 526) according to the alphabetical order.
147) Please define abbreviation for "FTLD" in "FTLD-tau" in Table 1.
148) Please replace "isoform / filament" with "isoform/filament" (2x), "Pick’s disease /" with "Pick’s disease/", "p-Tau" with "p-tau" (2x), "regions / phenotype" with "regions/phenotype" (2x), "Frontal/temporal degeneration" with "FTLD" in Table 1.
149) Please remove bold formatting from "Aberrant N-glycosylation", "Enrichment of high-mannose N-glycans", "Relative enrichment of complex-type N-glycans", "Abnormal Tau glycosylation", "Reduced O-GlcNAcylation", "less reduced", "O-GlcNAc deficits also observed" in Table 1.
150) Please provide reference for "By preserving polysialylation, sialidase inhibitors may protect synaptic plasticity" (line 531).
151) Please change "Process / Target" to "Process/target", "Type of glycosylation" to "Type of glycosylation", "Biomarker / Therapeutic Relevance" to "Biomarker/therapeutic relevance", "Key References" to "Key references" in Table 2.
152) Please define abbreviation for "AD" in the legend to Table 2.
153) Please replace "evidences demonstrate" with "evidence demonstrates" (line 545).
154) Please change "Glycosylation" to something like "To this end, glycosylation" (line 551).
155) "A diagram of a cell AI-generated content may be incorrect." sign appears when hovering the mouse cursor over Figure 2. Please disable this feature.
156) Please replace "Tau" with "tau" in Figure 2.
157) Please define abbreviation for "CDK5", "NFT" in the legend to Figure 2.
158) Please merge "GSK3β, glycogen synthase kinase 3 beta; IL-1β, Interleukin 1β; IL-6, Interleukin 6; OGA, O-GlcNAcase; OGT, O-GlcNAc transferase; O-GlcNAc, O-linked N-acetylglucosamine; N-glycosylation, asparagine-linked glycosylation; PHF, paired helical filament; TNF-α, tumour necrosis factor α." (line 575) with the legend to Figure 2.
159) Please provide reference for "TREM2 requires N-glycosylation for stability" (line 580).
160) Please change "quantification, and" to "quantification and" (line 591).
161) Please merge "Aberrant N-glycosylation primes tau for hyperphosphorylation and aggregation while reduced O-GlcNAcylation removes a critical brake against pathogenic modifications [91,116]." (line 612) with the previous paragraph.
162) Please merge "N- and O-glycosylation of APP and its processing enzymes profoundly influence amyloidogenic cleavage, with downstream effects on Aβ burden [5,117]." (line 615) with the previous paragraph.
163) Please merge "Glycosylation of receptors and adhesion molecules determines synaptic resilience, while glycosylation of immune receptors (TREM2, CD33, and RAGE) amplifies maladaptive neuroinflammation [19,27,118]." (line 617) with the previous paragraph.
164) Please merge "Enzymes such as OGT, OGA, and glycosyltransferases orchestrate glycosylation balance. Their dysregulation, often linked to impaired glucose metabolism and HBP flux, 621 couples systemic metabolic dysfunction to neurodegeneration [3,119]." (line 620) with the previous paragraph.
165) Please provide reference for "Enzymes such as OGT, OGA, and glycosyltransferases orchestrate glycosylation balance" (line 620).
166) Please merge "Glycoproteomic studies reveal distinct glycosylation signatures in brain, CSF, and serum, offering diagnostic and prognostic markers and providing disease-specific fingerprints that differentiate AD from other tauopathies [11,34,120]." (line 623) with the previous paragraph.
167) Please replace "Glycosylation-Focused Guidelines for Tauopathies: Toward Consensus on Tau Pathology" with "Glycosylation-focused guidelines for tauopathies: toward consensus on tau pathology" (line 629).
168) Please sort "Asn, Asparagine; C-terminal, Carboxi-terminal region; GSKβ, Glycogen synthase kinase 3 beta; MARK2, Microtubule Affinity-Regulating Kinase 2; N-terminal, N-terminal region; N-glycosylation, N-linked glycosylation; O-GlcNAc, O-linked N-acetylglucosamine; OGA, O-GlcNAcase, OGT, O-GlcNAc Transferase; Ser, Serine; Thr, Threonine; 2N4R, Tau isoform with 2 N-terminal inserts and 4 microtubule-binding repeats" (line 637) according to the alphabetical order.
169) Please merge "Asn, Asparagine; C-terminal, Carboxi-terminal region; GSKβ, Glycogen synthase kinase 3 beta; MARK2, Microtubule Affinity-Regulating Kinase 2; N-terminal, N-terminal region; N-glycosylation, N-linked glycosylation; O-GlcNAc, O-linked N-acetylglucosamine; OGA, O-GlcNAcase, OGT, O-GlcNAc Transferase; Ser, Serine; Thr, Threonine; 2N4R, Tau isoform with 2 N-terminal inserts and 4 microtubule-binding repeats" (line 637) with the legend to Figure 3.
170) Please change "Crosstalk on Tau Protein" to "crosstalk on tau protein" (line 636).
171) From "Tau isoform context (3R, 4R, or mixed) and filament type (PHF vs. straight), with region(s) sampled" (line 643) is not unequivocally clear whether the authors mean to say "region" or "regions"? Please fix.
172) Please replace "p-Tau" with "phospho-tau" (lines 648, 673).
173) Please change "AD: mixed" to "Mixed" (line 653).
174) Please replace "PSP/CBD: predominantly" with "Predominantly" (line 655).
175) Please change "FTLD-tau subtypes: specify" to "Specify" (line 658).
176) Please replace "Phospho-Tau" with "Phospho-tau" (line 661).
177) Please change "IHC/CSF; always" to "IHC/CSF. Always" (line 661).
178) From "Glyco-Tau antibodies (N-glycan or O-GlcNAc-specific) remain research-grade; pair them with: (i) enzymatic toggling controls (PNGase F for N-glycans; OGA/OGT modulation for O-GlcNAc), (ii) lectin co-capture, and (iii) MS confirmation on the same samples—especially for CSF/serum where abundance is low" (line 663) is not unequivocally clear whether the authors mean to say "OGA and OGT" or "OGA or OGT"? Please correct.
179) Similarly, from "Glyco-Tau antibodies (N-glycan or O-GlcNAc-specific) remain research-grade; pair them with: (i) enzymatic toggling controls (PNGase F for N-glycans; OGA/OGT modulation for O-GlcNAc), (ii) lectin co-capture, and (iii) MS confirmation on the same samples—especially for CSF/serum where abundance is low" (line 663) is not unequivocally clear whether the authors mean to say "CSF and serum" or "CSF or serum"? Please fix.
180) Please replace "Glyco-Tau" with "Glyco-tau" (line 663).
181) Please change "research-grade; pair" to "research-grade. Pair" (line 663).
182) Please replace "fluids; for" with "fluids. For" (line 667).
183) From "Tier 1 (exploratory): site discovery in brain tissue (MS-centric) with matched phospho/O-GlcNAc data" (line 670) is not unequivocally clear whether the authors mean to say "phospho and O-GlcNAc" or "phospho or O-GlcNAc"? Please correct.
184) Please change "Tier 1 (exploratory): site" to "Site" (line 670).
185) Please replace "Tier 2 (verification): antibody–lectin" with "Antibody–lectin" (line 672).
186) Please change "CSF; report" to "CSF. Report" (line 672).
187) From "Tier 3 (qualification): multi-center CSF/plasma panels combining p-Tau with Tau glyco-epitopes + APP/immune glyco-markers for differential diagnosis (AD vs. PSP/CBD)" (line 673) is not unequivocally clear whether the authors mean to say "CSF and plasma panels" or "CSF or plasma panels"? Please fix.
188) Similarly, from "Tier 3 (qualification): multi-center CSF/plasma panels combining p-Tau with Tau glyco-epitopes + APP/immune glyco-markers for differential diagnosis (AD vs. PSP/CBD)" (line 673) is not unequivocally clear whether the authors mean to say "APP and immune glyco-markers" or "APP or immune glyco-markers"? Please fix.
189) Please replace "Tier 3 (qualification): multi-center" with "Multi-center" (line 673).
190) Please change "Interleukin" to "interleukin" (line 575 2x).
191) Please replace "Asparagine" with "asparagine" (line 637).
192) Please change "Carboxi-terminal" o "Carboxy-terminal" (line 637).
193) Please replace "Glycogen" with "glycogen" (line 637).
194) Please change "Microtubule Affinity-Regulating Kinase" to "microtubule affinity-regulating kinase" (line 638).
195) Please replace "Transferase" with "transferase" (line 640).
196) Please change "Serine" to "serine" (line 640).
197) Please replace "Threonine" with "threonine" (line 640).
198) Please provide reference for "High-resolution glycoproteomics using advanced mass spectrometry, lectin arrays, and nanoparticle probes should be applied to large, well-characterized cohorts" (line 683).
199) Please change "M.B. and" to "M.B., and" (line 756).
Author Response
Thank you for all your comments and suggestions. They aided in increasing the value of our manuscript.

Round 3
Reviewer 3 Report
Comments and Suggestions for Authors
Major points:
The manuscript still lacks general clarity in terms of way too many minor mistakes (see the list below). Please focus namely on duly referencing and table abbreviations.
Minor points:
1) Please change "(CBD), share" to "(CBD) share" (line 20).
2) Please provide reference for "Alzheimer’s disease (AD) is the leading cause of dementia affecting over 50 million individuals worldwide, with prevalence projected to rise sharply as populations age" (line 41).
3) Please provide reference for "A crucial layer of regulation in these processes is post-translational modification (PTM), which fine-tunes protein structure, stability, and localization" (line 49).
4) Please provide reference for "Glycosylation refers to the enzymatic addition of glycans to proteins or lipids" (line 52).
5) Please provide reference for "Variants such as O-GlcNAcylation—the dynamic attachment of N-acetylglucosamine—play pivotal roles in regulating protein interactions and preventing aggregation" (line 56).
6) Please provide reference for "Tauopathies, a group of neurodegenerative diseases characterized by abnormal aggregation of the microtubule-associated protein Tau include AD, progressive supranuclear palsy (PSP), corticobasal degeneration (CBD), and Pick’s disease" (line 60).
7) Please replace "Tau" with "tau" (lines 61, 166, 315).
8) Please provide reference for "In tauopathies such as frontotemporal lobar degeneration (FTLD), PSP, and CBD, glycosylation abnormalities intersect with tau phosphorylation, altering aggregation dynamics and neuronal toxicity" (line 65).
9) Please provide reference for "Similarly, amyloid precursor protein (APP) processing and Aβ secretion are influenced by N-glycosylation, linking glycan biology to amyloid pathology" (line 67).
10) Please provide reference for "Tau protein is a microtubule-associated protein essential for stabilizing cytoskeletal architecture and supporting axonal transport" (line 82).
11) Please provide reference for "Importantly, aberrant N-glycosylation often precedes or facilitates hyperphosphorylation" (line 95).
12) Please provide reference for "Glycosylated tau shows increased susceptibility to kinases such as glycogen synthase kinase-3β (GSK3β), leading to pathogenic phosphorylation pattern" (line 96).
13) Please provide reference for "Another critical modification is O-GlcNAcylation, the addition of O-linked N-acetylglucosamine to serine or threonine residues" (line 103).
14) Please change "hexosamine biosynthesis pathway (HBP)" to "HBP" (line 111).
15) Please provide reference for "In contrast, PSP and CBD predominantly accumulate 4R-tau, forming straight filaments and involving basal ganglia, brainstem, and motor cortex" (line 126).
16) Please justify the format of the text presented in the "2.5. Comparative analysis of tau glycosylation and phosphorylation sites" chapter (line 148).
17) Please provide reference for "N-glycosylation appears to promote tau hyperphosphorylation and aggregation, whereas O-GlcNAcylation exerts a protective role" (line 149).
18) Please remove bold formatting from "(" in "(CDK5)" (line 157).
19) Please provide reference for "N-glycosylation changes the tertiary structure of tau, reducing its microtubule affinity and exposing neighboring serine and threonine residues to kinases, thereby facilitating hyperphosphorylation" (line 159).
20) Please provide reference for "In contrast, O-GlcNAcylation directly competes with phosphorylation at the same or adjacent residues, inhibiting excessive phosphorylation and stabilizing tau–microtubule interactions" (line 161).
21) Please merge "In AD, both 3R and 4R Tau isoforms exhibit extensive N-glycosylation and reduced O-GlcNAcylation, driving PHF assembly. In other tauopathies such as PSP and CBD, tau aggregates are composed predominantly of 4R isoforms with more restricted glycosylation patterns and distinct phosphorylation site usage (e.g., Ser262, Ser396). These biochemical differences suggest that glycosylation and phosphorylation site occupancy contribute to the diversity of filament morphology and regional vulnerability observed 171 across tauopathies [10,11,13]." (line 166) with the preceding paragraph.
22) Please provide reference for "In AD, both 3R and 4R Tau isoforms exhibit extensive N-glycosylation and reduced O-GlcNAcylation, driving PHF assembly" (line 166).
23) Please provide reference for "In other tauopathies such as PSP and CBD, tau aggregates are composed predominantly of 4R isoforms with more restricted glycosylation patterns and distinct phosphorylation site usage (e.g., Ser262, Ser396)" (line 167).
24) Please merge "Future work should address these site-specific interactions, emphasizing that N-glycosylation (Asn167/359) acts as an early structural modifier that primes tau for pathogenic phosphorylation, while O-GlcNAcylation (Ser400, Thr245, Ser356) counteracts aggregation by masking phosphorylation motifs. This comparative framework could enhance understanding of tau pathology across AD and non-AD tauopathies and guide the design of site-directed therapeutic interventions." (line 173) with the preceding paragraph.
25) Please provide reference for "Future work should address these site-specific interactions, emphasizing that N-glycosylation (Asn167/359) acts as an early structural modifier that primes tau for pathogenic phosphorylation, while O-GlcNAcylation (Ser400, Thr245, Ser356) counteracts aggregation by masking phosphorylation motifs" (line 173).
26) Please provide reference for "APP is a heavily N-glycosylated type I transmembrane protein" (line 187).
27) Please provide reference for "In addition to N-glycosylation, APP and its secretases undergo O-glycosylation, particularly O-GalNAc modification, which influences protein trafficking and enzymatic activity" (line 198).
28) Please provide reference for "BACE1 contains multiple N-glycans that regulate folding, transport, and stability" (line 203).
29) Please provide reference for "Similarly, γ-secretase components such as nicastrin are highly glycosylated" (line 205).
30) Please provide reference for "The glycan structures near the substrate-binding pocket determine APP cleavage pattern" (line 206).
31) Please provide reference for "Emerging evidence suggests that Aβ peptides themselves can undergo glycosylation or interact with glycans" (line 210).
32) Please provide reference for "Modified Aβ shows altered aggregation kinetics and toxicity" (line 211).
33) Please provide reference for "Glycosylation also modulates Aβ clearance via receptor-mediated endocytosis" (line 215).
34) Please provide reference for "Furthermore, microglial and astrocytic receptors involved in Aβ clearance—such as triggering receptor expressed on myeloid cells 2 (TREM2) and cluster of differentiation 33 (CD33)—are heavily glycosylated" (line 227).
35) Please replace "amyloid-β" with "Aβ" (line 240).
36) Please provide reference for "Neuronal communication depends on the proper glycosylation of neurotransmitter receptors and cell adhesion proteins" (line 248).
37) Please provide reference for "NMDA receptors, critical for synaptic plasticity, require N-glycosylation for assembly, trafficking, and surface expression" (line 249).
38) Please provide reference for "Similarly, α-amino-3-hydroxy-5-methyl-4-isoxazolepropionic acid (AMPA) receptors rely on N-glycans for stability and gating properties" (line 253).
39) Please provide reference for "Adhesion molecules such as neural cell adhesion molecule (NCAM) and L1CAM are highly glycosylated, and polysialylation of NCAM regulates neurite outgrowth and synaptic remodeling" (line 257).
40) Please change "Reduced" to something "In addition, reduced" (line 259).
41) Please provide reference for "Microglia, the brain’s resident immune cells, rely on glycosylated receptors for recognition and clearance of pathological proteins" (line 262).
42) Please provide reference for "TREM2 contains multiple N-glycosylation sites that regulate folding and surface expression" (line 263).
43) Please provide reference for "Interleukin-6 (IL-6), a pro-inflammatory cytokine elevated in AD, requires N-glycosylation for secretion and receptor binding" (line 276).
44) Please provide reference for "The complement cascade, a major mediator of synaptic pruning and neuroinflammation, is highly influenced by glycosylation" (line 286).
45) Please replace "cerebrospinal fluid" with "CSF" (line 292).
46) Please provide reference for "In addition, toll-like receptors (TLRs) rely on N-glycans for stability and ligand recognition" (line 300).
47) Please define abbreviation for "IgG" (line 325), "Fc" (line 325).
48) Please provide reference for "While recent studies have employed antibody–lectin hybrid assays and antibody enrichment followed by LC–MS/MS to enhance specificity, no single antibody currently demonstrates broad cross-platform reliability" (line 360).
49) Please change "LC–MS/MS" to "liquid chromatography (LC)–MS/MS" (line 361).
50) Please replace "CNS proteins" with "CNS" (line 373).
51) Please change "p-tau181, p-tau217" to "phospho-tau181, phospho-tau217" (line 386).
52) Please change "quantification, and" to "quantification and" (line 390).
53) Please replace "biomarkers, and" with "biomarkers and" (line 394).
54) Please provide reference for "In addition, longitudinal studies are beginning to reveal how glycosylation changes track disease trajectory, paving the way for prognostic applications" (line 395).
55) Please replace "glycoproteomics" with "glycoproteomics." (lines 400, 401, 612, 614).
56) Please define abbreviation for "HILIC" in "Lectin/HILIC", "CSF" in "CSF/plasma", "FDR", "PTMs" in Table 1.
57) Please format "O" in "O-GlcNAc" using italics in Table 1.
58) Please format "N" in "de-N-glycosylation" using italics in Table 1.
59) Please change "Description / Technologies" to "Description/Technologies", "+" to "plus" in Table 2.
60) Please define abbreviation for "ETD", "EThcD", "AI-ETD", "HCD" in "stepped-HCD", "FAIMS", "TIMS-PASEF", "CSF", "PRM", "DIA-SWATH", "PTM", "IMS", "LC" in "LC-MS", "FDR" in Table 2.
61) Please replace "Alzheimer’s disease" with "AD" (lines 405, 444).
62) Please change "AD to "Alzheimer’s disease" (line 409).
63) Please provide reference for "N-acetylglucosaminyltransferases (GnTs) regulate branching and extension of N-glycans" (line 410).
64) Please provide reference for "Abnormal N-glycosylation of tau promotes hyperphosphorylation and aggregation, while reduced O-GlcNAcylation removes a protective brake on phosphorylation" (line 432).
65) Please provide reference for "N- and O-glycosylation of APP and secretases regulate Aβ generation, with aberrant patterns favoring amyloidogenic processing" (line 434).
66) Please provide reference for "Synaptic receptors and adhesion molecules require proper glycosylation for stability and plasticity" (line 435).
67) Please provide reference for "Their dysregulation contributes to synaptic failure" (line 437).
68) Please provide reference for "Immune receptors (TREM2, CD33, and RAGE) and cytokines are glycosylated, shaping microglial and astrocytic activation, while complement protein glycosylation regulates synapse pruning" (line 437).
69) Please provide reference for "Finally, altered activity of glycosyltransferases, glycosidases, and the HBP underlies these shifts" (line 439).
70) "A diagram of a cell cycle AI-generated content may be incorrect." sign appears when hovering the mouse cursor over Figure 1. Please disable this feature.
71) Please provide reference for "HBP provides UDP-N-acetylglucosamine (UDP-GlcNAc), the substrate for O-GlcNAcylation" (line 451).
72) Please provide reference for "Experimental restoration of HBP flux via glucosamine supplementation increases O-GlcNAcylation and reduces tau phosphorylation in animal models, highlighting its therapeutic potential" (line 456).
73) Please provide reference for "Aberrant expression of glycosylation enzymes is detectable in CSF and serum, offering biomarker potential" (line 471).
74) Please justify the format of the text presented in the "6.6. Clinical correlations of tau glycosylation with cognitive decline and disease staging" chapter.
75) Please merge "Conversely, reduced O-GlcNAcylation levels in tau correlate with accelerated cognitive decline and lower Mini-Mental State Examination (MMSE) scores, suggesting a loss of neuroprotective regulation. Longitudinal cohort studies have shown that specific CSF glycoforms of tau—particularly those carrying high-mannose N-glycans or lacking O-GlcNAc at Ser400 and Thr245—predict progression from mild cognitive impairment (MCI) to Alzheimer’s dementia over 2–4 years [76,82]." (line 482) with the preceding paragraph.
76) From "In AD, elevated N-glycosylated tau in brain tissue and CSF has been associated with advanced Braak stages and greater neurofibrillary tangle burden, reflecting a molecular shift toward aggregation-prone conformers" (line 479) is not clear what the authors mean by "Braak stages"? Please provide a simple definition.
77) Please provide reference for "Conversely, reduced O-GlcNAcylation levels in tau correlate with accelerated cognitive decline and lower Mini-Mental State Examination (MMSE) scores, suggesting a loss of neuroprotective regulation" (line 482).
78) Please merge "Furthermore, glyco-profiling in CSF and plasma using lectin-based assays or glycoproteomic MS has demonstrated potential for distinguishing AD from other tauopathies such as PSP and CBD, based on differences in glycan branching and sialylation patterns, as shown in Table 3. Collectively, these findings suggest that the glycosylation state of tau serves as a dynamic biochemical correlate of disease stage and may complement phosphorylated tau and Aβ42 as part of a next-generation biomarker panel for early detection and disease monitoring [83]." (line 488) with the preceding paragraph.
79) Please provide reference for ""Furthermore, glyco-profiling in CSF and plasma using lectin-based assays or glycoproteomic MS has demonstrated potential for distinguishing AD from other tauopathies such as PSP and CBD, based on differences in glycan branching and sialylation patterns" mentioned in "Furthermore, glyco-profiling in CSF and plasma using lectin-based assays or glycoproteomic MS has demonstrated potential for distinguishing AD from other tauopathies such as PSP and CBD, based on differences in glycan branching and sialylation patterns, as shown in Table 3" (line 488).
80) Please define abbreviation for "AD", "CBD", "PSP", "p-tau" in the legend to Table 3.
81) Please change "p-tau" to "phospho-tau" in Table 3 (2x).
82) Please replace "PHF, paired helical filament" with "PHFs, paired helical filaments" (line 497).
83) Please define abbreviation for "BACE1" in the legend to Table 4.
84) Please change "Balance" to "Balances" in Table 4.
85) Please replace "Alzheimer’s disease (AD)" with "AD" (line 516).
86) "A diagram of a cell AI-generated content may be incorrect." sign appears when hovering the mouse cursor over Figure 2. Please disable this feature.
87) Please change "phosphorylation" to "phosphorylation," (line 545).
88) Please replace "Kinase" with "kinase" (line 547).
89) Please change "tumour" to "tumor" (line 550).
90) Please provide reference for "Glycosylation critically modulates immune receptor signaling" (line 552).
91) Please provide reference for "REM2 requires N-glycosylation for stability" (line 553).
92) Please format "Created with BioRender." (line 605) using font size consistent with the rest of the text.
93) Please replace "Carboxy-terminal" with "carboxy-terminal" (line 607).
94) Please change "3 beta" to "3-β" (line 607).
95) Please format "N" in "N-acetylglucosamine" using italics" (line 609).
96) Please define abbreviation for "AD", "APP" in "APP-related", "CP13", "CSF", "ELISA", "IHC", "IP" in "IP→LC-MS/MS", "LC" in "IP→LC-MS/MS", "LoD", "LoQ", "PHF", "PHF-1", "PNGase", "PTM", "PSP" in "PSP/CBD", "CBD" in "PSP/CBD" in Table 5.
97) Please format "O" in "O-GlcNAc" using italics in Table 5 (5x).
98) Please format "N" in "N-glycosylation" using italics in Table 5 (2x).
99) Please format "N" in "N-glycan" using italics in Table 5.
100) "8. Future directions" section seems to be fragmented into many small paragraphs. Please merge these into larger but coherent structures.
101) Please provide reference for "High-resolution glycoproteomics using advanced mass spectrometry, lectin arrays, and nanoparticle probes should be applied to large, well-characterized cohorts" (line 622).
102) Please replace "mass spectrometry" with "MS" (line 622).
103) Please provide reference for "Clinical translation of OGA inhibitors and other glycosylation-modulating compounds should be pursued cautiously with attention to the dualistic nature of glycosylation, protective vs. pathogenic" (line 626).
Author Response
Thank you for all your comments and suggestions. We have made all the mofifications accordingly.
